- 1 Measurement Report: Collocated speciation and potential
- 2 mechanisms of gaseous adsorption for integrated filter-based
- 3 sampling and analysis of water-soluble organic molecular
- 4 markers in the atmosphere

- 7 Wei Feng<sup>1</sup>, Xiangyu Zhang<sup>1</sup>, Zhijuan Shao<sup>2</sup>, Guofeng Shen<sup>3</sup>, Hong Liao<sup>1</sup>, Yuhang
- 8 Wang<sup>4</sup>, Mingjie Xie<sup>1,\*</sup>

9

10

- 11 <sup>1</sup>Collaborative Innovation Center of Atmospheric Environment and Equipment
- 12 Technology, Jiangsu Key Laboratory of Atmospheric Environment Monitoring and
- 13 Pollution Control, School of Environmental Science and Engineering, Nanjing
- 14 University of Information Science & Technology, 219 Ningliu Road, Nanjing 210044,
- 15 China.
- <sup>2</sup>School of Environment Science and Engineering, Suzhou University of Science and
- 17 Technology Shihu Campus, 99 Xuefu Road, Suzhou 215009, China
- <sup>3</sup>Laboratory for Earth Surface Processes, College of Urban and Environmental Sciences,
- 19 Peking University, Beijing 100871, China
- <sup>4</sup>School of Earth and Atmospheric Sciences, Georgia Institute of Technology, Atlanta
- 21 GA 30332, United States


- 23 \*Correspondence to:
- 24 Mingjie Xie (mingjie.xie@nuist.edu.cn, mingjie.xie@colorado.edu);
- Mailing address: 219 Ningliu Road, Nanjing, Jiangsu, 210044, China



#### Abstract






















To better understand the measurement uncertainties and sampling artifacts of particulate water-soluble organic molecular markers (WSOMMs), three quartz filters were stacked and installed in two collocated samplers (Sampler I and II) to simultaneously collect ambient WSOMMs. The paired top filters  $(Q_t)$  loaded with PM<sub>2.5</sub> were analyzed to determine the duplicate-derived uncertainty of particulate WSOMM concentrations. For several WSOMMs (e.g., levoglucosan) specifically associated with aerosol sources, the uncertainty was well below 20%, which was commonly assumed in previous studies for the analysis of particulate WSOMMs. If the WSOMMs detected in the other two filters ( $Q_b$  and  $Q_{bb}$ ) below  $Q_f$  were caused by gaseous adsorption, the breakthrough value ( $[Q_{bb}]/([Q_b]+[Q_{bb}])$ ) can be used to estimate the sampling artifact of particulate WSOMMs due to gaseous adsorption on Q<sub>f</sub>. To understand the influence of acidic and alkaline conditions on the adsorption of gaseous WSOMMs or their precursors on quartz filters, the bottom filter (Qbb) of Sampler I was treated with (NH<sub>4</sub>)<sub>2</sub>SO<sub>4</sub> or KOH on different sampling days. From the comparison of the measurement results between chemically treated and untreated Qbb samples, it was inferred that (NH<sub>4</sub>)<sub>2</sub>SO<sub>4</sub> can increase the formation of isoprene secondary organic aerosol by reactive uptake of the oxidative intermediates; KOH can promote the adsorption of organic acids through neutralization reactions. Future studies are warranted to develop a suitable method for sampling gaseous WSOMMs using chemically treated adsorbents.




#### 1. Introduction


























As a major component of atmospheric aerosols, water-soluble organic carbon (WSOC) can influence aerosol radiative forcing through absorbing and scattering solar and terrestrial radiation (Malm et al., 1996; Ming et al., 2005) and promoting cloud formation by acting as cloud condensation nuclei and ice-nucleating particles (Novakov and Penner, 1993; Chen et al., 2021). Moreover, the deposition of WSOC provides nutrients for plants and microorganisms on Earth that maintain the balance of the ecosystem (Quinn et al., 2010; Iavorivska et al., 2017; Goll et al., 2023). The heavy metals and toxic organics associated with WSOC also increase the health risks of atmospheric aerosols (Tao and Lin, 2000). WSOC can be released directly by biomass burning (Ding et al., 2013; Du et al., 2014) or can be formed by the atmospheric oxidation of volatile organic precursors and subsequent gas-particle partitioning processes (termed "secondary organic aerosol", SOA) (Zhang et al., 2007; Kroll and Seinfeld, 2008). Water-soluble organic molecular makers (WSOMMs) are organic compounds with specific origins in the atmosphere and are commonly used to identify the sources of WSOC and particulate matter (PM). In laboratory studies where SOA formation was simulated using a smoke chamber, WSOMMs play a central role in revealing the reaction pathways (Kroll et al., 2006; Ng et al., 2008). A comprehensive understanding of the physicochemical properties, atmospheric transformation and environmental impacts of WSOC depends largely on its characterization (Noziere et al., 2015). Uncertainty analysis for the quantification of PM components, including WSOC and WSOMMs, is necessary to show the variability of measurement results due to sampling, pretreatment, instrumental analysis, etc (Zhang et al., 2024). The uncertainty data are also needed when the simulation results of atmospheric transport models, e.g. for predicting the spatiotemporal distribution of PM components and SOA formation, are evaluated by comparison with measurements (Aleksankina et al., 2019). For PM species with high measurement uncertainty, modeling could aim to obtain a reasonable range instead of a specific value. In addition, the uncertainty data are required for source apportionment using receptor models (Kim and Hopke, 2007). In existing studies, propagation methods (e.g., root sum of squares) have been used to predict the overall uncertainty of the system from different sources of uncertainty (Jaeckels et al., 2007; Dutton et al., 2009b; Feng et al., 2023b). Another method to estimate the uncertainty is to conduct repeated analysis for selected samples (Xie et al., 2016), which only considers the error during chemical analysis. The total uncertainty for the characterization of atmospheric composition is composed of the uncertainties in both sampling and chemical analysis, and can be directly determined by performing collocated sampling. This method has been applied to estimate the concentration uncertainties of bulk PM components (Dutton et al., 2009a; Yang et al., 2021; Xie et al., 2022b), but the duplicate-derived uncertainty for the characterization of WSOMMs has rarely been investigated. The known WSOMMs (e.g., 2-methyltetrols) are mostly semi-volatile organic compounds (SVOCs), in which a mass transfer always takes place between the gas and particle phase (Yatavelli et al., 2014; Xie et al., 2014b). In filter-based sampling of WSOMMs in the particle phase, the adsorption of gaseous WSOMMs on filters ("blow on" effect, positive artifact) leads to an overestimation of particle-phase concentrations (Hart and Pankow, 1994; Mader and Pankow, 2001b; Subramanian et al., 2004). Several studies have used a denuder to eliminate organic gasses in the air stream prior to sampling PM on filters (Eatough et al., 2003; Fan et al., 2004; Subramanian et al., 2004), which creates a large potential for volatilization ("blow off" effect, negative artifact) of particulate organic matter (OM) due to the disruption of the gas-particle equilibrium

























(Subramanian et al., 2004; Watson et al., 2009). The use of a backup quartz filter downstream of the PM-loaded quartz filter or a Teflon filter has been used in many studies to correct for adsorption of gaseous organics, with the target species being mostly bulk organic carbon (OC) (Watson and Chow, 2002; Subramanian et al., 2004; 2009) and non-polar organic compounds (e.g., *n*-alkanes and polycyclic aromatics) (Mader and Pankow, 2001a; Xie et al., 2014a), while sampling artifacts of WSOMMs were less considered.


























The existence of gaseous WSOMMs has been reported by integrated gas-particle (G-P) sampling (Limbeck et al., 2005; Bao et al., 2012; Liu et al., 2012; Shen et al., 2018) or online measurements (Williams et al., 2010; Xu et al., 2019; Lv et al., 2022a, 2022b). Polyurethane foam (PUF) was the most commonly used adsorbent for sampling gaseous WSOMMs in offline observations (Xie et al., 2014b; Shen et al., 2020; Lanzafame et al., 2021; Qin et al., 2021). However, the extraction process could be affected by the leaching of the PUF material in methanol, leading to low recoveries (approximately 50%). To prove that methacrylic acid epoxide (MAE) is the key intermediate for the formation of 2-methylglyceric acid (2-MG) from isoprene under high NO<sub>X</sub> conditions, Lin et al. (2013b) collected gaseous MAE using an ice-cooled glass bubbler filled with ethyl acetate. Due to the limited flow rate and absorption efficiency, this liquid absorption method was more suitable for qualitative rather than quantitative purposes. The Semi-Volatile Thermal Desorption Aerosol Gas chromatograph (SV-TAG) was developed for hourly measurements of WSOMMs in the gas and particle phase. In the SV-TAG, a parallel thermal desorption cell equipped with passivated high-surface-area stainless steel (SS) fiber filters (F-CTD) was used for sampling (Williams et al., 2010; Zhao et al., 2013a, 2013b; Isaacman et al., 2014, 2016). One F-CTD was used to directly collect WSOMMs in both the gas and particle phases, while the other cell was set up to collect only WSOMMs in the particle phase by passing the sample air through an upstream activated carbon denuder. Comparisons between the two cells directly reflected the G-P partitioning of the WSOMMs. However, the resulting particulate fraction (F%) was often greater than 100% (Isaacman et al., 2016; Liang et al., 2023), possibly due to the uncertainties associated with the small sampling volume and chemical analysis.

In this study, three quartz filters were stacked and installed in two collocated samplers for sampling WSOMMs. The measurement results of WSOMMs on the top filter were used to estimate the uncertainties of analyzing WSOMMs in the particle phase. The remaining two bare quartz filters in one sampler were analyzed to assess positive sampling artifacts due to adsorption of gaseous WSOMMs or their precursors. To investigate the impacts of acidic and alkaline conditions on the adsorption on quartz filters, the bottom filter of the other sampler was soaked in ammonium sulfate ((NH<sub>4</sub>)<sub>2</sub>SO<sub>4</sub>) or potassium hydroxide (KOH) and dried before sampling. The study results unveil the uncertainties in the characterization of WSOMMs in the particle phase, and are beneficial for further studies on sampling and analysis of gaseous WSOMMs.

#### 2. Methods

#### 2.1 Sampling

All filter samples were collected on the rooftop of a six-story building (Binjiang Building) of Nanjing University of Information Science and Technology (NUIST, 32.21°N, 118.71°E). The sampling site is located in a suburb in the western Yangtze River Delta of China (Figure 1a), approximately 20 km north of the city center of Nanjing. The inter-provincial highway G40 and Jiangbei expressway are located about 700 m and 1.5 km northwest and southeast, respectively. The petrochemical industry of Yangzi and the chemical industry of Nanjing (SINOPEC) are located 5 – 10 km

northeast of the site. The surrounding area consists mainly of residential buildings, road traffic, and parks (e.g., the Longwangshan scenic area).

Three quartz filters (20.3 cm × 12.6 cm, Munktell Filter AB, Sweden) were stacked and placed on each of the two identical samplers (Sampler I and II; Mingve Environmental, Guangzhou, China) equipped with 2.5 µm cut impactors to collect ambient air at a flow rate of 300 L min<sup>-1</sup>, with a filter face velocity of 25.2 cm S<sup>-1</sup>. All filters were pre-baked at 550°C for 4 h to remove potential organic contaminants. Twenty-four pairs of collocated samples were collected from August to September 2021 during daytime (08:00 - 19:00 GMT + 8, N = 12) and nighttime (20:00 - 07:00 the next)day, GMT+8, N=12). As shown in Figure 1b, the top filter ( $Q_f$ ) in each filter was loaded with PM<sub>2.5</sub>, and the subsequent two filters ( $Q_b$  and  $Q_{bb}$ ) were used to evaluate the adsorption of gaseous WSOMMs or their precursors on filters. In Sampler I, Qbb was soaked in 1 M (NH<sub>4</sub>)<sub>2</sub>SO<sub>4</sub> (N = 12) or 1 M KOH (N = 12) and dried at a temperature of 120°C before sampling, while Q<sub>bb</sub> in Sampler II was not treated with chemicals. Table S1 shows the sampling date, mean temperature and relative humidity (RH, %), and the type of Q<sub>bb</sub> treatment ((NH<sub>4</sub>)<sub>2</sub>SO<sub>4</sub>, KOH, and no treatment) of Sampler I and II. Field blanks were taken at every 10<sup>th</sup> sample to correct for possible contamination. All samples and field blanks were sealed and stored at -20°C until analysis.

#### 2.2 Chemical analysis


























The method of analysis for WSOMMs in filter samples has been detailly described in our previous work (Qin et al., 2021; Feng et al., 2023a, 2023b). Briefly, one-eighth of each filter sample was spiked with 40  $\mu$ L of deuterated internal standards (IS, succinic acid-d4, levoglucosan-d7, naphthalene-d8, acenaphthene-d10, phenanthrene-d10, chrysene-d10, and perylene-d12; 10 ng  $\mu$ L<sup>-1</sup>) and ultrasonically extracted twice for 15 min in a mixture of methanol and dichloromethane (v:v, 1:1). The total extract

of each sample was then rotary evaporated and blown to dryness with a gentle stream of  $N_2$ . 60  $\mu L$  of derivatization reagent [N, O-bis(trimethylsilyl)trifluoroacetamide (BSTFA) with 1% trimethylchlorosilane (TMCS) and pyridine, 5:1] was added and reacted with the dried extracts at 70°C for 3 hours. Prior to instrumental analysis by gas chromatography (GC, Agilent 7890B, USA)-mass spectrometry (MS, Agilent-5977B, USA), the extract solution was cooled to room temperature and diluted with 340  $\mu L$  of pure hexane. Quantification of the individual WSOMMs was performed by generating six-point calibration curves and the IS method.

Water-soluble inorganic ions and WSOC in filter samples were extracted with ultrapure water (18.2 M $\Omega$ ). Cations (NH<sub>4</sub><sup>+</sup>, K<sup>+</sup>, Ca<sup>2+</sup> and Mg<sup>2+</sup>) and anions (SO<sub>4</sub><sup>2-</sup> and NO<sub>3</sub><sup>-</sup>) were determined using Metrohm (930, Switzerland) and Dionex (ICS-3000, USA) ion chromatography (IC), respectively. WSOC was analyzed using a total organic carbon analyzer (TOC-L, Shimadzu, Japan). Bulk OC and elemental carbon (EC) of the filter samples were measured using a thermal-optical carbon analyzer (DRI, 2001A, Atmoslytic, USA) according to the IMPROVE-A protocol. Field blanks were analyzed in the same way as the air samples, and the measurement results of all filter samples were corrected.

2.3 Data analysis

2.3.1 Breakthrough calculation

When  $Q_b$  and  $Q_{bb}$  were considered as adsorbents for sampling gaseous WSOMMs, the WSOMM concentrations in the  $Q_b$  and  $Q_{bb}$  samples can be used to calculate the breakthrough (B), which represents the sampling efficiency and is defined as follows:

$$B = \frac{[Q_{bb}]}{[Q_b] + [Q_{bb}]} \times 100\% \tag{1}$$

where  $[Q_b]$  and  $[Q_{bb}]$  represent the concentrations of each target compound in  $Q_b$  and  $Q_{bb}$  samples, respectively. A *B* value of 33% has been commonly used as a threshold

- for excessive breakthrough, and a *B* value of close to or higher than 50% indicates complete breakthrough (Peters et al., 2000).
- 2.3.2 Calculation of the fractions of particulate and adsorbed WSOMMs
- Assuming that the target WSOMMs measured in the  $Q_f$  samples exist in the particle phase, and those detected in the  $Q_b$  and  $Q_{bb}$  samples are present in the gas phase, the particulate (F%) and adsorption (A%) fractions of the individual WSOMMs can be calculated as follows:

$$F\% = \frac{[Q_f]}{[Q_f] + [Q_{bb}]} \times 100\%$$
 (2)

$$212 A\% = 1 - F\% (3)$$

- where  $[Q_f]$  denotes the concentrations of the target compound in  $Q_f$  samples.
- 2.14 2.3.3 Uncertainty assessment
- The coefficient of divergence (COD) has often been used as a measure of the similarity of chemical species concentrations between pairs of PM samples (Wilson et al., 2005) and is defined as follows:

$$COD = \sqrt{\frac{1}{n} \sum_{i=1}^{n} (\frac{x_{i1} - x_{i2}}{x_{i1} + x_{i2}})^2}$$
 (4)

- where  $x_{il}$  and  $x_{i2}$  in this work are the concentrations of a particular WSOMM in the i<sup>th</sup> pair of  $Q_f$  samples from Sampler I and Sampler II, respectively, and n is the number of sample pairs. Values of COD approaching 0 and 1 indicate identity and complete divergence between pairs of collocated samples.
- The standard deviation of paired differences (SD<sub>diff</sub>) and average relative percent difference (ARPD) were used to quantify the absolute and relative uncertainties of individual WSOMMs based on collocated measurement data (Flanagan et al., 2006; Dutton et al., 2009a; Yang et al., 2021). They were calculated as follows:

SD<sub>diff</sub> = 
$$\sqrt{\frac{1}{2n}\sum_{i=1}^{n}(x_{i1}-x_{i2})^2}$$
 (5)

$$ARPD = \frac{2}{n} \sum_{i=1}^{n} \frac{|x_{i1} - x_{i2}|}{(x_{i1} + x_{i2})} \times 100\%$$
 (6)

#### 229 3 Results and discussion

- 3.1 Collocated measurements of  $Q_f$  samples
- 3.1.1 Overview of the measurement data
- The mean concentrations of WSOMMs and bulk PM<sub>2.5</sub> components in collocated 232 Q<sub>f</sub> samples are summarized in Tables 1 and S2, respectively. Generally, all species 233 showed similar mean concentrations between paired  $Q_f$  samples with no significant 234 difference (Student's t test, p = 0.55 - 0.96). Among the isoprene SOA tracers, the mean 235 concentration of 2-methylglyceric acid (2-MG,  $4.48 \pm 3.15$  ng m<sup>-3</sup>) was comparable to 236 observations at the same site in summer 2019 (3.62  $\pm$  1.38 ng m<sup>-3</sup>) and summer 2020 237  $(4.71 \pm 1.77 \text{ ng m}^{-3})$  (Feng et al., 2023b). However, the mean concentrations of 2-238 methyltetrols (2-MTs,  $13.1 \pm 7.00$  ng m<sup>-3</sup>) and C<sub>5</sub>-alkene triols (C<sub>5</sub>-ATs,  $15.6 \pm 14.7$  ng 239  $m^{-3}$ ) were significantly (p < 0.01) lower than in summer 2019 (21.3 ± 18.2 ng  $m^{-3}$ , 21.3 240  $\pm$  26.9 ng m<sup>-3</sup>) and summer 2020 (27.0  $\pm$  21.6 ng m<sup>-3</sup>, 36.3  $\pm$  48.0 ng m<sup>-3</sup>). After the 241 242 implementation of a series of air pollution control measures in China after 2013 (e.g., 243 the "Air Pollution Prevention and Control Action Plan"), an annual decrease in sulfate concentrations was observed in Nanjing (Xie et al., 2022a). As shown in Table S2, the 244 mean sulfate concentration in this study  $(5.82 \pm 2.07 \text{ ng m}^{-3})$  is lower than in summer 245  $2019 (8.92 \pm 3.25 \text{ ng m}^{-3})$  and summer  $2020 (7.67 \pm 2.92 \text{ ng m}^{-3})$  (Feng et al., 2023b). 246 247 Since sulfate participates in the reactive uptake of isoprene SOA intermediates to form C<sub>5</sub>-ATs, 2-MTs, and hydroxy sulfate esters (Surratt et al., 2007a, 2010), the decrease in 248 249 sulfate concentrations is a possible reason for the attenuation of isoprene SOA formation (Worton et al., 2013; Lin et al., 2013a; Xu et al., 2015). The concentrations 250 of the primary WSOMMs, including biomass burning tracers, saccharides, and sugar 251 alcohols, in this study had similar mean concentrations as in summer 2019 and summer 252

2020 (Feng et al., 2023a). This could be due to the weak emissions from biomass burning in summer with little annual variation (Zhang et al., 2008; Li et al., 2020; Fu et al., 2023), and sugar polyols mainly originate from biogenic sources during the growing season with minimal influence from human activities (Simoneit et al., 2004; Jia and Fraser, 2011; Kang et al., 2018).

#### 3.1.2 Duplicate-derived uncertainty


























Figures 2 and S1 show comparisons of the concentrations of selected typical WSOMMs and other compounds in collocated  $Q_f$  samples. The scattering data of all identified WSOMMs fell along the identity line with strong correlations (r > 0.90, p <0.01). The COD values of all species were below 0.20, indicating a high similarity between the collocated measurements (Krudysz et al., 2008). Yang et al. (2021) found that the median concentrations of bulk PM<sub>2.5</sub> components were negatively correlated with the corresponding ARPD values. In this work, such dependence of measurement uncertainties on ambient concentration was not observed for WSOMMs, possibly due to the high sensitivity of GC-MS analysis for derivatized WSOMMs. The SD<sub>diff</sub> and ARPD values shown in Figures 2 and S1 are the uncertainties for particulate WSOMMs based on direct measurements, which are rarely reported. When using measurement data of particulate WSOMMs for receptor-based source apportionment (e.g., positive matrix factorization), uncertainty data are a required input and are often estimated using a propagation method (Hemann et al., 2009; Dutton et al., 2009a; Aleksankina et al., 2019), where an error fraction of 20% was usually assumed (Zhang et al., 2009). However, the ARPD values of several WSOMMs (e.g., levoglucosan, 2-MTH and mannosan) specifically related to PM sources were close to or even below 10% (Figures 2 and S1), and overestimation of uncertainties may lead to biased source apportionment results (Paatero and Hopke, 2003).

In previous studies, meso-erythritol was often used as a surrogate for the quantification of all isoprene SOA tracers (Ding et al., 2008; Hu et al., 2008; Lin et al., 2012; Feng et al., 2023b). Due to differences in molecular structures, MS fragments, and signal intensities, quantification of target compounds using surrogates can be subject to errors. As shown in Figure S2a and c, the quantification results of 2-MG and 2-MEH using authentic standards and meso-erythritol (surrogate) are strongly correlated (r = 0.99, p < 0.01). But the mean concentration of 2-MG quantified using the authentic standard was 14.9% higher than that using the surrogate (Figure S2 b). The difference in the quantification of 2-MEH between using the authentic and surrogate standards was not apparent, which was attributed to the similarity of the structure of meso-erythrol and 2-MEH. To obtain more accurate measurement results of WSOMMs, authentic standards or at least surrogates with similar structures should be used for quantification.

3.2 Adsorption of gaseous WSOMMs or their precursors on untreated filters

Owning to the extremely low vapor pressures of the biomass burning tracers, saccharides, and sugar alcohols (Qin et al., 2021), these species were not detected in the  $Q_b$  and  $Q_{bb}$  samples from Sampler II or showed similar concentrations as the field blanks. Therefore, only the measurement results of isoprene SOA tracers and dicarboxylic acids in  $Q_b$  and  $Q_{bb}$  samples are presented and discussed. When  $Q_b$  and  $Q_{bb}$  were considered as adsorbents for sampling gaseous WSOMMs, the mean F% values of isoprene SOA tracers and dicarboxylic acids are well above 50% (Table 2). However, significant amounts of the target species were observed in the  $Q_b$  and  $Q_{bb}$  samples, indicating that the quartz filter can adsorb semi-volatile WSOMMs in the gas phase or their precursors that undergo heterogeneous reactions at the filter surface. After the sampling air flowed through  $Q_f$  and  $Q_b$  of Sampler II, the vapor pressures of the

target compounds or precursors decreased significantly, resulting in lower concentrations in  $Q_{bb}$  samples than in  $Q_b$  samples (Table 2).


























Qin et al. (2021) collected particulate and gaseous WSOMMs at the same observation site by passing air samples through stacked  $Q_f$  and  $Q_b$  and a PUF plug. Similar to this study,  $Q_f$  was used to determine the particulate WSOMMs. Assuming that the WSOMMs detected in filters and PUF after  $Q_f$  are present in the gas phase, Figure 3 compares the concentrations of isoprene SOA tracers in different sampling matrices of this study and Qin et al. (2021) during the same period (August – September) of the year. In Figure 3a, the mean  $Q_f$  and  $Q_b$  concentrations of 2-MTs (13.5  $\pm$  7.16 ng  $m^{-3}$  and 1.61  $\pm$  1.53 ng  $m^{-3}$ ; Table 2) and C<sub>5</sub>-ATs (16.0  $\pm$  14.7 ng  $m^{-3}$ , 0.24  $\pm$  0.13 ng  $m^{-3}$ ) are lower in this study than in Qin et al. (2021) (2-MTs 20.7 ± 17.6 ng  $m^{-3}$ , 3.96 ± 5.41 ng m<sup>-3</sup>; C<sub>5</sub>-ATs  $22.0 \pm 26.5$  ng m<sup>-3</sup>,  $1.18 \pm 1.42$  ng m<sup>-3</sup>). However, the Q<sub>bb</sub> samples in this study had comparable or even higher mean concentrations of 2-MTs (0.75  $\pm$  0.87 ng m<sup>-3</sup>) and C<sub>5</sub>-ATs (0.23  $\pm$  0.29 ng m<sup>-3</sup>) than the PUF samples (2-MTs 0.99  $\pm$  0.75 ng  $m^{-3}$ ; C<sub>5</sub>-ATs 0.065 ± 0.062 ng  $m^{-3}$ ; Figure 3c). Figure S3 shows that the F% of 2-MTs and C<sub>5</sub>-ATs (2-MTs  $86.3 \pm 8.90 \text{ ng m}^{-3}$ , C<sub>5</sub>-ATs  $94.9 \pm 5.80 \text{ ng m}^{-3}$ ) are similar in this study and in Qin et al. (2021) (2-MTs  $81.8 \pm 9.85$  ng m<sup>-3</sup>, C<sub>5</sub>-ATs  $91.7 \pm 7.69$  ng m<sup>-3</sup>) under comparable meteorological conditions, although the sampling year and sampling media are different. Thus, there is no appreciable difference in the gas-particle partitioning results between the use of quartz filters and PUF for sampling isoprene SOA tracers in the gas phase. Based on PM<sub>2.5</sub> data obtained from a nearby monitoring station using the same method as (Yu et al., 2019), the mean PM<sub>2.5</sub> concentration during the sampling period in (Qin et al., 2021) (28.2  $\pm$  10.3 µg m<sup>-3</sup>) was significantly (p <0.05) higher than in this study (15.3  $\pm$  5.29  $\mu$ g m<sup>-3</sup>; Figure S3b), indicating that particle loading may not be a major factor affecting the G-P partitioning of isoprene SOA tracers.

Considering the higher recoveries in the measurement of isoprene SOA tracers in filter samples ( $106 \pm 1.90\%$ ) than in PUF samples (about 50%), which are largely due to the elution of PUF materials, quartz filters can be used instead of PUF for sampling. The SV-TAG method proposes that SS fiber filters are suitable for sampling SVOCs in the gas phase if their surface area is large enough (Zhao et al., 2013b). The specific fiber surface area of quartz filters (~130 cm<sup>2</sup> cm<sup>-2</sup>) is slightly lower than that of SS fiber filters (~160 cm<sup>2</sup> cm<sup>-2</sup>) (Mader and Pankow, 2001b; Zhao et al., 2013b), but the diameter of quartz filters (≥90 mm) used for ambient sampling can be much larger. Without considering the heterogeneous reactions on the filter surfaces, no excessive breakthrough (B < 33%) was observed for 2-MG and 2-MTs based on the measurement results of the Q<sub>b</sub> and Q<sub>bb</sub> samples from Sampler II, but the B values of C<sub>5</sub>-ATs and dicarboxylic acids are close to 50% (complete breakthrough). These results suggest that bare quartz filters are not effective adsorbents for sampling C5-ATs and dicarboxylic acids in the gas phase. According to the equilibrium G-P partitioning theory, a greater fraction of SVOCs exists in the gas phase when temperature rises (Pankow, 1994a, 1994b), as the vapor pressure of SVOCs increases exponentially with temperature. More adsorption sites on filter surfaces can be blocked by H<sub>2</sub>O molecules with increased RH, leading to lower adsorption of SVOCs (Pankow et al., 1993). Since absorption by particulate organic matter (OM) is an important G-P partitioning mechanism for ambient SVOCs (Liang and Pankow, 1996; Liang et al., 1997), increased OC concentrations might correspond to higher particle-phase fractions of SVOCs. However, the A% values of isoprene SOA tracers and dicarboxylic acids show little dependence on temperature, RH, and OC concentrations (Figures S4–S6), indicating more complex mechanisms for the adsorption of WSOMMs than for nonpolar SVOCs (e.g., n-alkanes and PAHs). For example, due to the hygroscopicity of

























WSOMMs, water molecules attached to filter surfaces can promote gaseous adsorption. Dissolution in aerosol liquid water is more important than absorption by particulate OM for the equilibrium between particle- and gas-phase WSOMMs (Kampf et al., 2013; Isaacman et al., 2016; Shen et al., 2018; Qin et al., 2021). Although the mean A\% values of dicarboxylic acids increased with their subcooled liquid vapor pressure  $(p^{o,*}_{L}; Figure$ S7), this dependence was not observed when isoprene SOA tracers were included, which is assumed to result from their formation through heterogeneous reactions on filter surfaces. Since adsorption of gaseous WSOMMs on quartz filters is a potential source of artifacts when sampling particulate WSOMMs (Arhami et al., 2006), previous studies have adjusted the particulate concentrations of organic compounds by subtracting the amounts on  $Q_b$  samples from those on  $Q_f$  samples ( $[Q_f]$ – $[Q_b]$ ) (Mader and Pankow, 2000, 2001a, 2001b). In this approach, the amounts of gaseous organic compounds adsorbed in  $Q_f$  and  $Q_b$  samples are assumed to be equal, and evaporation of the particle phase is neglected. However,  $Q_b$  is exposed to lower concentrations of gaseous WSOMMs before  $Q_f$  reaches equilibrium with the air sample (Mader and Pankow, 2001b; Watson et al., 2009). Then, the  $[Q_f]-[Q_h]$  method may lead to an overestimation of particulate concentrations unless the sampling time is long enough (Hart and Pankow, 1994; Subramanian et al., 2004). In Sampler II of this study, a third bare quartz filter  $(Q_{bb})$  was added after  $Q_f$  and  $Q_b$ , and the B values given in Table 2 also reflect the relationship between the amounts of gaseous WSOMMs adsorbed on two consecutive quartz filters. As such, it is more appropriate to estimate the amounts of gaseous WSOMMs adsorbed on  $Q_f([Q_f^*])$  by assuming that the B value of  $Q_f$  and  $Q_b$  ( $[Q_b]/([Q_f^*]+[Q_b])$ ) is identical to that of  $Q_b$  and


























Q<sub>bb</sub>. In this case, the artifact-corrected particulate concentrations of the WSOMMs can

be calculated as  $[Q_f]$ – $[Q_f^*]$ . As Figure 4 shows, the  $[Q_f]$ ,  $[Q_f]$ – $[Q_b]$ , and  $[Q_f]$ – $[Q_f^*]$  values of all six species have similar time series. However, except for C<sub>5</sub>-ATs, the mean  $[Q_f]$  and  $[Q_f]$ – $[Q_b]$  values of 2-MG, 2-MTs, and dicarboxylic acids are 33.8% – 78.1% and 11.1% – 40.3% higher than that of  $[Q_f]$ – $[Q_f^*]$ . Since the volatilization of particulate WSOMMs in  $Q_f$  samples was not known, the values of  $[Q_f]$ – $[Q_f^*]$  can be regarded as a lower limit for filter-based measurements of particulate WSOMMs.

#### 3.3 Adsorption of gaseous WSOMMs or their precursors on treated filters

The sampling efficiency of gaseous WSOMMs can be improved by treating the sampling medium with chemicals. Bao et al. (2012) collected gaseous organic acids using two tandem annular denuders coated with potassium hydroxide (KOH), and obtained a sampling efficiency up to 98% for short-chain dicarboxylic acids ( $C_2 - C_6$ ). Kawamura and Kaplan (1987) and Bock et al. (2017) used KOH-impregnated quartz filters to collect motor vehicle emissions, and confirmed that engine exhaust is a source of dicarboxylic acids. In this study, the  $Q_{bb}$  on Sampler I was treated with (NH<sub>4</sub>)<sub>2</sub>SO<sub>4</sub> or KOH on different sampling days (Table S1). Table 3 compares the measurement results of the  $Q_b$  and (NH<sub>4</sub>)<sub>2</sub>SO<sub>4</sub>-treated  $Q_{bb}$  samples from Sampler I with those of the collocated samples from Sampler II. The mean concentrations of 2-MTs and  $C_5$ -ATs in the treated  $Q_{bb}$  samples from Sampler I were  $3.34 \pm 2.64$  ng m<sup>-3</sup> and  $3.92 \pm 3.25$  ng m<sup>-3</sup>, respectively, which were 2.83 and 22.1 times higher than those in the untreated  $Q_{bb}$  samples from Sampler II. While the collocated  $Q_b$  samples had similar mean concentrations of 2-MTs and  $C_5$ -ATs.

Referring to the results of the chamber study, 2-MTs and C<sub>5</sub>-ATs are formed by the reactive uptake of epoxydiols of isoprene (IEPOX) through the acid-catalyzed ring opening (Surratt et al., 2006, 2010). The coated (NH<sub>4</sub>)<sub>2</sub>SO<sub>4</sub> on Q<sub>bb</sub> can absorb water vapor and act as an acid to promote the hydrolysis of IEPOX on filters to form 2-MTs

and C<sub>5</sub>-ATs. In addition, inorganic sulfate on filters can also react with gaseous IEPOX as a nucleophile to form organosulfate esters and oligomeric forms of 2-MTs and C<sub>5</sub>-ATs. As shown in Table S2,  $Q_b$  and untreated  $Q_{bb}$  samples from Sampler II also contain a certain amount of inorganic sulfate due to the heterogeneous reactions of SO<sub>2</sub> (Pierson et al., 1980; Cheng et al., 2012), which are favored by the reactive uptake of IEPOX. The concentrations of  $SO_4^{2-}$  and  $NH_4^+$  in the  $Q_b$  samples from Sampler I ( $SO_4^{2-}$  0.13  $\pm$  $0.056~\mu g~m^{-3};~NH_4^+~0.033~\pm~0.026~\mu g~m^{-3})$  and II  $(0.10\pm0.040~\mu g~m^{-3},~0.024\pm0.022$ μg m<sup>-3</sup>) were comparable, indicating that there was no significant transfer of (NH<sub>4</sub>)<sub>2</sub>SO<sub>4</sub> from treated  $Q_{bb}$  to  $Q_b$  on Sampler I during sampling. This also explains the similar concentrations of 2-MTs and C<sub>5</sub>-ATs in Q<sub>b</sub> samples between Sampler I and II. During the derivatization process of sample analysis, the organosulfate and oligomeric forms of 2-MTs and C5-ATs can be converted to their monomeric forms by excess BSTFA (Lin et al., 2013a; Xie et al., 2014b); the conventional GC/EI-MS method also overestimates the concentrations of 2-MTs and C5-ATs due to the thermal decomposition of less volatile oligomers and organosulfates (Lopez et al., 2016; Cui et al., 2018). Consequently, 2-MTs and C<sub>5</sub>-ATs detected in the Q<sub>b</sub> and Q<sub>bb</sub> samples from both Sampler I and II were likely generated by heterogeneous reactions of gaseous IEPOX on quartz filter surfaces rather than by direct adsorption of gaseous molecules. Unlike 2-MTs and C<sub>5</sub>-ATs, 2-MG in  $(NH_4)_2SO_4$ -treated  $Q_{bb}$  samples  $(0.16 \pm 0.12)_4$ ng m<sup>-3</sup>; Table 3) did no show higher mean concentration in comparison to that in untreated  $Q_{bb}$  samples (0.24  $\pm$  0.16 ng m<sup>-3</sup>). 2-MG is formed by the acid-catalyzed ring opening of MAE, an oxidation product of isoprene under high NO<sub>X</sub> conditions (Lin et al., 2013b). Surratt et al. (2007b) demonstrated that the formation of 2-MG is almost unaffected by changes in the acidity of the aerosol. Thus, 2-MG is stable in acidic aerosols and an equilibrium between the gas and particle phase could be achieved. The

























mean concentrations of succinic acid, glutaric acid, and adipic acid in  $(NH_4)_2SO_4$ -treated  $Q_{bb}$  samples were 11.3%, 57.4%, and 74.1% higher, respectively, than those in untreated  $Q_{bb}$  samples (Table 3). One possible explanation is that  $(NH_4)_2SO_4$  is highly hygroscopic and promotes the dissolution of gaseous dicarboxylic acids by moisture absorption (Chen et al., 2021) or facilitates the heterogeneous formation of dicarboxylic acids (Yli et al., 2013; Bikkina et al., 2017).

Table 4 shows that the mean concentrations of 2-MG ( $1.92 \pm 1.38 \text{ ng m}^{-3}$ ), succinic acid  $(7.05 \pm 5.39 \text{ ng m}^{-3})$ , glutaric acid  $(1.50 \pm 1.71 \text{ ng m}^{-3})$ , and adipic acid  $(1.16 \pm$ 1.20 ng m<sup>-3</sup>) in KOH-treated Q<sub>bb</sub> samples from Sampler I are up to 13.7 times higher than those in untreated  $Q_{bb}$  samples from Sampler II. This can be explained by the formation of low-volatility organic compounds by neutralization reactions of gaseous organic acids on the surface of KOH-treated Qbb. As described in section 3.2, the breakthrough in the sampling of gaseous 2-MG (24.1  $\pm$  10.2%) and 2-MTs (28.1  $\pm$ 13.1%) is not excessively high when bare quartz filters are used. However, their concentrations in KOH- and (NH<sub>4</sub>)<sub>2</sub>SO<sub>4</sub>-treated Q<sub>bb</sub> samples increased substantially compared to untreated Q<sub>bb</sub> samples (Tables 3 and 4), indicating that a low B value does not guarantee high sampling efficiency of gaseous WSOMMs or their precursors. Owing to the transfer of KOH from treated  $Q_{bb}$  to  $Q_b$  on Sampler I, the mean concentrations of 2-MG and dicarboxylic acids in Q<sub>b</sub> samples from Sampler I are 1.84 - 2.26 times higher than those in Sampler II (Table 4). The reactive uptake of organic acids in Q<sub>b</sub> samples from Sampler I during KOH treatment periods also led to increased WSOC and OC concentrations, and the transferred KOH on Q<sub>b</sub> accelerated the heterogeneous formation of SO<sub>4</sub><sup>2</sup>- and NO<sub>3</sub>- (Table S2).

### 4. Implications and conclusions


























In this study, the uncertainties for the concentrations of particulate WSOMMs (5.85%

| $-19.9\%$ ) were determined by direct measurements of collocated $Q_f$ samples. The                                               |
|-----------------------------------------------------------------------------------------------------------------------------------|
| uncertainties for several compounds (e.g., levoglucosan and 2-MTH) were well below                                                |
| the default value (~20%) commonly used in previous studies. The uncertainty data                                                  |
| presented in this work are useful for future modeling and field studies on atmospheric                                            |
| transport, transformation, and source apportionment of water-soluble organic aerosols.                                            |
| When the bare $Q_b$ and $Q_{bb}$ are considered as adsorbents for sampling gas-phase                                              |
| WSOMMs, the $F\%$ values obtained in this study are comparable to those obtained at                                               |
| the same sampling site using PUF as adsorbent. Based on the breakthrough of gaseous                                               |
| isoprene SOA tracers and dicarboxylic acids calculated from the measurement results                                               |
| of $Q_b$ and $Q_{bb}$ samples, a new method was developed to correct for the adsorption of                                        |
| gaseous organics on PM-loaded filter samples (Qf), which accounts for the decrease in                                             |
| gas-phase concentrations after the air sample passes through Qf. The adjusted Qf                                                  |
| measurements could be used as a lower limit for the particulate concentrations of                                                 |
| WSOMMs.                                                                                                                           |
| By comparing the concentrations of isoprene SOA tracers and dicarboxylic acids                                                    |
| between (NH <sub>4</sub> ) <sub>2</sub> SO <sub>4</sub> -/KOH-treated and untreated Q <sub>bb</sub> samples, it was inferred that |
| $(NH_4)_2SO_4$ on quartz filters can promote the heterogeneous formation of 2-MTs and $C_5$ -                                     |
| ATs by reactive uptake of IEPOX, and KOH can increase the adsorption of gaseous                                                   |
| organic acids on quartz filters by neutralization reactions. Due to the influence of                                              |
| surface reactions, WSOMMs detected in adsorbents associated with SOA sources (e.g.,                                               |
| 2-MTs) may not indicate their existence in the gas phase. In further studies, chemically                                          |
| treated adsorbents can be developed for sampling gaseous WSOMMs with specific                                                     |

### Data Availability

functional groups.

| 478                      | Data used in the writing of this paper (and its Supplementary Information file)                                                                                                                                                                                                                                         |  |  |  |  |  |  |  |
|--------------------------|-------------------------------------------------------------------------------------------------------------------------------------------------------------------------------------------------------------------------------------------------------------------------------------------------------------------------|--|--|--|--|--|--|--|
| 479                      | are publicly available on Harvard Dataverse (Feng et al., 2025,                                                                                                                                                                                                                                                         |  |  |  |  |  |  |  |
| 480                      | https://doi.org/10.7910/DVN/ZD0JQW).                                                                                                                                                                                                                                                                                    |  |  |  |  |  |  |  |
| 481                      |                                                                                                                                                                                                                                                                                                                         |  |  |  |  |  |  |  |
| 482                      | Author contributions                                                                                                                                                                                                                                                                                                    |  |  |  |  |  |  |  |
| 483                      | MX designed the research. WF and XZ managed the sampling work and                                                                                                                                                                                                                                                       |  |  |  |  |  |  |  |
| 484                      | performed laboratory experiments. WF, XZ, and MX analyzed the data. WF and MX                                                                                                                                                                                                                                           |  |  |  |  |  |  |  |
| 485                      | wrote the paper with significant contributions from ZS, GS, HL, and YW.                                                                                                                                                                                                                                                 |  |  |  |  |  |  |  |
| 486                      |                                                                                                                                                                                                                                                                                                                         |  |  |  |  |  |  |  |
| 487                      | Competing interests                                                                                                                                                                                                                                                                                                     |  |  |  |  |  |  |  |
| 488                      | The contact author has declared that none of the authors has any competing                                                                                                                                                                                                                                              |  |  |  |  |  |  |  |
| 489                      | interests.                                                                                                                                                                                                                                                                                                              |  |  |  |  |  |  |  |
| 490                      |                                                                                                                                                                                                                                                                                                                         |  |  |  |  |  |  |  |
| 491                      | Acknowledgements                                                                                                                                                                                                                                                                                                        |  |  |  |  |  |  |  |
| 492                      | This work was supported by the National Natural Science Foundation of China                                                                                                                                                                                                                                             |  |  |  |  |  |  |  |
| 493                      | (NSFC, 42177211).                                                                                                                                                                                                                                                                                                       |  |  |  |  |  |  |  |
| 494                      |                                                                                                                                                                                                                                                                                                                         |  |  |  |  |  |  |  |
| 495                      |                                                                                                                                                                                                                                                                                                                         |  |  |  |  |  |  |  |
| 496                      | References                                                                                                                                                                                                                                                                                                              |  |  |  |  |  |  |  |
| 497<br>498<br>499        | Aleksankina, K., Reis, S., Vieno, M., and Heal, M. R.: Advanced methods for uncertainty assessment and global sensitivity analysis of an Eulerian atmospheric chemistry transport model, Atmos. Chem. Phys., 19(5), 2881-2898, https://doi.org/10.5194/acp-19-2881-2019, 2019.                                          |  |  |  |  |  |  |  |
| 500<br>501<br>502        | Arhami, M., Kuhn, T., Fine, P. M., Delfino, R. J., and Sioutas, C.: Effects of sampling artifacts and operating parameters on the performance of a semicontinuous particulate elemental carbon/organic carbon monitor, Environ. Sci. Technol., 40(3), 945-954,                                                          |  |  |  |  |  |  |  |
| 503<br>504<br>505<br>506 | https://doi.org/10.1021/es0510313, 2006.  Bao, L., Matsumoto, M., Kubota, T., Sekiguchi, K., Wang, Q., and Sakamoto, K.: Gas/particle partitioning of low-molecular-weight dicarboxylic acids at a suburban site in Saitama, Japan, Atmos. Environ., 47, 546-553, https://doi.org/10.1016/j.atmosenv.2009.09.014, 2012. |  |  |  |  |  |  |  |
| 507<br>508               | Bikkina, S., Kawamura, K., and Sarin, M.: Secondary organic aerosol formation over coastal ocean: Inferences from atmospheric water-soluble low molecular weight organic compounds, Environ.                                                                                                                            |  |  |  |  |  |  |  |
| 509<br>510               | Sci. Technol., 51(8), 4347-4357, https://doi.org/10.1021/acs.est.6b05986, 2017. Bock, N., Baum, M. M., Anderson, M. B., Pesta, A., and Northrop, W. F.: Dicarboxylic acid emissions                                                                                                                                     |  |  |  |  |  |  |  |

- from aftertreatment equipped diesel engines, Environ. Sci. Technol., 51(21), 13036-13043, https://doi.org/10.1021/acs.est.7b03868, 2017.
- Carlton, A., and Turpin, B.: Particle partitioning potential of organic compounds is highest in the Eastern US and driven by anthropogenic water, Atmos. Chem. Phys., 13(20), 10203-10214, https://doi.org/10.5194/acp-13-10203-2013, 2013.
- Chen, J., Wu, Z., Wu, G., Gong, X., Wang, F., Chen, J., Shi, G., Hu, M., and Cong, Z.: Ice-nucleating particle concentrations and sources in rainwater over the third pole, Tibetan Plateau, J. Geophys.
  Res.:Atmos., 126(9), e2020JD033864, https://doi.org/10.1029/2020JD033864, 2021.
  Chen, Y., Guo, H., Nah, T., Tanner, D. J., Sullivan, A. P., Takeuchi, M., Gao, Z., Vasilakos, P., Russell, A.

- Chen, Y., Guo, H., Nah, T., Tanner, D. J., Sullivan, A. P., Takeuchi, M., Gao, Z., Vasilakos, P., Russell, A. G., and Baumann, K.: Low-molecular-weight carboxylic acids in the Southeastern US: Formation, partitioning, and implications for organic aerosol aging, Environ. Sci. Technol., 55(10), 6688-6699, https://doi.org/10.1021/acs.est.1c01413, 2021.
- Cheng, Y., Duan, F., He, K., Du, Z., Zheng, M., and Ma, Y.: Sampling artifacts of organic and inorganic aerosol: Implications for the speciation measurement of particulate matter, Atmos. Environ., 55, 229-233, https://doi.org/10.1016/j.atmosenv.2012.03.032, 2012.
- Cui, T., Zeng, Z., Dos Santos, E. O., Zhang, Z., Chen, Y., Zhang, Y., Rose, C. A., Budisulistiorini, S. H., Collins, L. B., and Bodnar, W. M.: Development of a hydrophilic interaction liquid chromatography (HILIC) method for the chemical characterization of water-soluble isoprene epoxydiol (IEPOX)-derived secondary organic aerosol, Environ. Sci. Proc. Imp., 20(11), 1524-1536, https://doi.org/10.1039/C8EM00308D, 2018.
- Ding, X., Wang, X., Xie, Z., Zhang, Z., and Sun, L.: Impacts of Siberian biomass burning on organic aerosols over the North Pacific Ocean and the Arctic: Primary and secondary organic tracers, Environ. Sci. Technol., 47(7), 3149-3157, https://doi.org/10.1021/es3037093, 2013.
- Ding, X., Zheng, M., Yu, L., Zhang, X., Weber, R. J., Yan, B., Russell, A. G., Edgerton, E. S., and Wang, X.: Spatial and seasonal trends in biogenic secondary organic aerosol tracers and water-soluble organic carbon in the southeastern United States, Environ. Sci. Technol., 42(14), 5171-5176, https://doi.org/10.1021/es7032636, 2008.
- Du, Z., He, K., Cheng, Y., Duan, F., Ma, Y., Liu, J., Zhang, X., Zheng, M., and Weber, R.: A yearlong study of water-soluble organic carbon in Beijing I: Sources and its primary vs. secondary nature, Atmos. Environ., 92, 514-521, https://doi.org/10.1016/j.atmosenv.2014.04.060, 2014.
- Dutton, S. J., Schauer, J. J., Vedal, S., and Hannigan, M. P.: PM<sub>2.5</sub> characterization for time series studies: Pointwise uncertainty estimation and bulk speciation methods applied in Denver, Atmos. Environ., 43(5), 1136-1146, https://doi.org/10.1016/j.atmosenv.2008.10.003, 2009a.
- Dutton, S. J., Williams, D. E., Garcia, J. K., Vedal, S., and Hannigan, M. P.: PM<sub>2.5</sub> characterization for time series studies: Organic molecular marker speciation methods and observations from daily measurements in Denver, Atmos. Environ., 43(12), 2018-2030, https://doi.org/10.1016/j.atmosenv.2009.01.003, 2009b.
- Eatough, D. J., Long, R. W., Modey, W. K., and Eatough, N. L.: Semi-volatile secondary organic aerosol in urban atmospheres: meeting a measurement challenge, Atmos. Environ., 37(9-10), 1277-1292, https://doi.org/10.1016/S1352-2310(02)01020-8, 2003.
- Fan, X., Lee, P. K., Brook, J. R., and Mabury, S. A.: Improved measurement of seasonal and diurnal differences in the carbonaceous components of urban particulate matter using a denuder-based air sampler, Aerosol Sci. Technol., 38(S2), 63-69, https://doi.org/10.1080/027868290504090, 2004.
- Feng, W., Shao, Z., Wang, Q. g., and Xie, M.: Size-resolved light-absorbing organic carbon and organic molecular markers in Nanjing, east China: Seasonal variations and sources, Environ. Pollut., 332, 122006, https://doi.org/10.1016/j.envpol.2023.122006, 2023a.
- Feng, W., Wang, X., Shao, Z., Liao, H., Wang, Y., and Xie, M.: Time-resolved measurements of PM<sub>2.5</sub> chemical composition and brown carbon absorption in Nanjing, East China: Diurnal variations and organic tracer-based PMF analysis, J. Geophys. Res.:Atmos., 128(18), e2023JD039092, https://doi.org/10.1029/2023JD039092, 2023b.
- Feng, W., Zhang, X., Shao, Z., Shen, G., Liao, H, Wang, Y., and Xie, M.: Replication Data for: Collocated speciation and potential mechanisms of gaseous adsorption for integrated filter-based sampling and analysis of water-soluble organic molecular markers in the atmosphere (Version 2) [Dataset], Harvard Dataverse, https://doi.org/10.7910/DVN/ZD0JQW, 2025.
- Flanagan, J. B., Jayanty, R., Rickman, J., Edward E, and Peterson, M. R.: PM<sub>2.5</sub> Speciation Trends
  Network: Evaluation of whole-system uncertainties using data from sites with collocated
  samplers, J. Air Waste Manage. Assoc., 56(4), 492-499,
  https://doi.org/10.1080/10473289.2006.10464516, 2006.
- Fu, M., Li, H., Wang, L., Tian, M., Qin, X., Zou, X., Chen, C., Wang, G., Deng, C., and Huang, K.:

- Atmospheric saccharides over the East China Sea: Assessment of the contribution of sea-land emission and the aging of levoglucosan, Sci. Total. Environ., 898, 165328, https://doi.org/10.1016/j.scitotenv.2023.165328, 2023.
- Goll, D. S., Bauters, M., Zhang, H., Ciais, P., Balkanski, Y., Wang, R., and Verbeeck, H.: Atmospheric phosphorus deposition amplifies carbon sinks in simulations of a tropical forest in Central Africa, New Phytol., 237(6), 2054-2068, https://doi.org/10.1111/nph.18535, 2023.

- Hart, K. M., and Pankow, J. F.: High-volume air sampler for particle and gas sampling. 2. Use of backup filters to correct for the adsorption of gas-phase polycyclic aromatic hydrocarbons to the front filter, Environ. Sci. Technol., 28(4), 655-661, https://doi.org/10.1021/es00053a019, 1994.
- Hemann, J., Brinkman, G., Dutton, S., Hannigan, M., Milford, J., and Miller, S.: Assessing positive matrix factorization model fit: a new method to estimate uncertainty and bias in factor contributions at the measurement time scale, Atmos. Chem. Phys., 9(2), 497-513, https://doi.org/10.5194/acp-9-497-2009, 2009.
- Hu, D., Bian, Q., Li, T. W., Lau, A. K., and Yu, J. Z.: Contributions of isoprene, monoterpenes, β-caryophyllene, and toluene to secondary organic aerosols in Hong Kong during the summer of 2006, J. Geophys. Res.:Atmos., 113(D22), D22206, https://doi.org/10.1029/2008JD010437, 2008.
- Iavorivska, L., Boyer, E. W., and Grimm, J. W.: Wet atmospheric deposition of organic carbon: An underreported source of carbon to watersheds in the northeastern United States, J. Geophys. Res.:Atmos., 122(5), 3104-3115, https://doi.org/10.1002/2016JD026027, 2017.
- Isaacman, V., G., Kreisberg, N., Yee, L., Worton, D., Chan, A., Moss, J., Hering, S., and Goldstein, A.: Online derivatization for hourly measurements of gas-and particle-phase semi-volatile oxygenated organic compounds by thermal desorption aerosol gas chromatography (SV-TAG), Atmos. Meas. Tech., 7(12), 4417-4429, https://doi.org/10.5194/amt-7-4417-2014, 2014.
- Isaacman, V., G., Yee, L. D., Kreisberg, N. M., Wernis, R., Moss, J. A., Hering, S. V., de Sa, S. S., Martin, S. T., Alexander, M. L., Palm, B. B., Hu, W., Campuzano-Jost, P., Day, D. A., Jimenez, J. L., Riva, M., Surratt, J. D., Viegas, J., Manzi, A., Edgerton, E., Baumann, K., Souza, R., Artaxo, P., and Goldstein, A. H.: Ambient Gas-Particle Partitioning of Tracers for Biogenic Oxidation, Environ. Sci. Technol., 50(18), 9952-9962, https://doi.org/10.1021/acs.est.6b01674, 2016.
- Jaeckels, J. M., Bae, M.-S., and Schauer, J. J.: Positive matrix factorization (PMF) analysis of molecular marker measurements to quantify the sources of organic aerosols, Environ. Sci. Technol., 41(16), 5763-5769, https://doi.org/10.1021/es062536b, 2007.
- Jia, Y., and Fraser, M.: Characterization of saccharides in size-fractionated ambient particulate matter and aerosol sources: the contribution of primary biological aerosol particles (PBAPs) and soil to ambient particulate matter, Environ. Sci. Technol., 45(3), 930-936, https://doi.org/10.1021/es103104e, 2011.
- Kampf, C. J., Waxman, E. M., Slowik, J. G., Dommen, J., Pfaffenberger, L., Praplan, A. P., Prevot, A. S., Baltensperger, U., Hoffmann, T., and Volkamer, R.: Effective Henry's law partitioning and the salting constant of glyoxal in aerosols containing sulfate, Environ. Sci. Technol., 47(9), 4236-4244, https://doi.org/10.1021/es400083d, 2013.
- Kang, M., Ren, L., Ren, H., Zhao, Y., Kawamura, K., Zhang, H., Wei, L., Sun, Y., Wang, Z., and Fu, P.: Primary biogenic and anthropogenic sources of organic aerosols in Beijing, China: Insights from saccharides and n-alkanes, Environ. Pollut., 243, 1579-1587, https://doi.org/10.1016/j.envpol.2018.09.118, 2018.
- Kawamura, K., and Kaplan, I. R.: Motor exhaust emissions as a primary source for dicarboxylic acids in Los Angeles ambient air, Environ. Sci. Technol., 21(1), 105-110, https://doi.org/10.1021/es00155a014, 1987.
- Kim, E., and Hopke, P. K.: Comparison between sample-species specific uncertainties and estimated uncertainties for the source apportionment of the speciation trends network data, Atmos. Environ., 41(3), 567-575, https://doi.org/10.1016/j.atmosenv.2006.08.023, 2007.
- Kroll, J. H., Ng, N. L., Murphy, S. M., Flagan, R. C., and Seinfeld, J. H.: Secondary organic aerosol formation from isoprene photooxidation, Environ. Sci. Technol., 40(6), 1869-1877, https://doi.org/10.1021/es0524301, 2006.
- Kroll, J. H., and Seinfeld, J. H.: Chemistry of secondary organic aerosol: Formation and evolution of low-volatility organics in the atmosphere, Atmos. Environ., 42(16), 3593-3624, https://doi.org/10.1016/j.atmosenv.2008.01.003, 2008.
- Krudysz, M. A., Froines, J. R., Fine, P. M., and Sioutas, C.: Intra-community spatial variation of sizefractionated PM mass, OC, EC, and trace elements in the Long Beach, CA area, Atmos. Environ., 42(21), 5374-5389, https://doi.org/10.1016/j.atmosenv.2008.02.060, 2008.
- Lanzafame, G., Srivastava, D., Favez, O., Bandowe, B., Shahpoury, P., Lammel, G., Bonnaire, N.,

Alleman, L., Couvidat, F., and Bessagnet, B.: One-year measurements of secondary organic aerosol (SOA) markers in the Paris region (France): Concentrations, gas/particle partitioning and SOA source apportionment, Sci. Total. Environ., 757, 143921, https://doi.org/10.1016/j.scitotenv.2020.143921, 2021.

- Li, W., Wang, M., Chen, M., Hu, K., Ge, X., Nie, D., Gu, C., Yu, W., and Cheng, Y.: Carbohydrates observations in suburb Nanjing, Yangtze River of Delta during 2017–2018: Concentration, seasonal variation, and source apportionment, Atmos. Environ., 243, 117843, https://doi.org/10.1016/j.atmosenv.2020.117843, 2020.
  - Liang, C., and Pankow, J. F.: Gas/particle partitioning of organic compounds to environmental tobacco smoke: partition coefficient measurements by desorption and comparison to urban particulate material, Environ. Sci. Technol., 30(9), 2800-2805, https://doi.org/10.1021/es960050x, 1996.
  - Liang, C., Pankow, J. F., Odum, J. R., and Seinfeld, J. H.: Gas/particle partitioning of semivolatile organic compounds to model inorganic, organic, and ambient smog aerosols, Environ. Sci. Technol., 31(11), 3086-3092, https://doi.org/10.1021/es9702529, 1997.
  - Liang, Y., Wernis, R. A., Kristensen, K., Kreisberg, N. M., Croteau, P. L., Herndon, S. C., Chan, A. W., Ng, N. L., and Goldstein, A. H.: Gas-particle partitioning of semivolatile organic compounds when wildfire smoke comes to town, Atmos. Chem. Phys., 23(19), 12441-12454, https://doi.org/10.5194/acp-23-12441-2023, 2023.
  - Limbeck, A., Kraxner, Y., and Puxbaum, H.: Gas to particle distribution of low molecular weight dicarboxylic acids at two different sites in central Europe (Austria), J. Aerosol Sci., 36(8), 991-1005, https://doi.org/10.1016/j.jaerosci.2004.11.013, 2005.
  - Lin, Y.-H., Knipping, E., Edgerton, E., Shaw, S., and Surratt, J.: Investigating the influences of SO<sub>2</sub> and NH<sub>3</sub> levels on isoprene-derived secondary organic aerosol formation using conditional sampling approaches, Atmos. Chem. Phys., 13(16), 8457-8470, https://doi.org/10.5194/acp-13-8457-2013, 2013a.
  - Lin, Y.-H., Zhang, H., Pye, H. O., Zhang, Z., Marth, W. J., Park, S., Arashiro, M., Cui, T., Budisulistiorini, S. H., and Sexton, K. G.: Epoxide as a precursor to secondary organic aerosol formation from isoprene photooxidation in the presence of nitrogen oxides, Proc. Natl. Acad. Sci. U.S.A., 110(17), 6718-6723, https://doi.org/10.1073/pnas.1221150110, 2013b.
  - Lin, Y.-H., Zhang, Z., Docherty, K. S., Zhang, H., Budisulistiorini, S. H., Rubitschun, C. L., Shaw, S. L., Knipping, E. M., Edgerton, E. S., and Kleindienst, T. E.: Isoprene epoxydiols as precursors to secondary organic aerosol formation: acid-catalyzed reactive uptake studies with authentic compounds, Environ. Sci. Technol., 46(1), 250-258, https://doi.org/10.1021/es202554c, 2012.
  - Liu, J., Zhang, X., Parker, E. T., Veres, P. R., Roberts, J. M., de Gouw, J. A., Hayes, P. L., Jimenez, J. L., Murphy, J. G., and Ellis, R. A.: On the gas-particle partitioning of soluble organic aerosol in two urban atmospheres with contrasting emissions: 2. Gas and particle phase formic acid, J. Geophys. Res.:Atmos., 117(D21), D00V21, https://doi.org/10.1029/2012JD017912, 2012.
  - Lopez, H., FD, Mohr, C., D'ambro, E., Lutz, A., Riedel, T., Gaston, C., Iyer, S., Zhang, Z., Gold, A., and Surratt, J.: Molecular composition and volatility of organic aerosol in the Southeastern US: implications for IEPOX derived SOA, Environ. Sci. Technol., 50(5), 2200-2209, https://doi.org/10.1021/acs.est.5b04769, 2016.
  - Lv, S., Wang, F., Wu, C., Chen, Y., Liu, S., Zhang, S., Li, D., Du, W., Zhang, F., and Wang, H.: Gas-to-aerosol phase partitioning of atmospheric water-soluble organic compounds at a rural site in China: An enhancing effect of NH<sub>3</sub> on SOA formation, Environ. Sci. Technol., 56(7), 3915-3924, https://doi.org/10.1021/acs.est.1c06855, 2022a.
  - Lv, S., Wu, C., Wang, F., Liu, X., Zhang, S., Chen, Y., Zhang, F., Yang, Y., Wang, H., and Huang, C.: Nitrate-enhanced gas-to-particle-phase partitioning of water-soluble organic compounds in Chinese urban atmosphere: Implications for secondary organic aerosol formation, Environ. Sci. Technol. Lett., 10(1), 14-20, https://doi.org/10.1021/acs.estlett.2c00894, 2022b.
  - Mader, B. T., and Pankow, J. F.: Gas/solid partitioning of semivolatile organic compounds (SOCs) to air filters. 1. Partitioning of polychlorinated dibenzodioxins, polychlorinated dibenzofurans and polycyclic aromatic hydrocarbons to teflon membrane filters, Atmos. Environ., 34(28), 4879-4887, https://doi.org/10.1016/S1352-2310(00)00241-7, 2000.
  - Mader, B. T., and Pankow, J. F.: Gas/solid partitioning of semivolatile organic compounds (SOCs) to air filters. 2. Partitioning of polychlorinated dibenzodioxins, polychlorinated dibenzofurans, and polycyclic aromatic hydrocarbons to quartz fiber filters, Atmos. Environ., 35(7), 1217-1223, https://doi.org/10.1016/S1352-2310(00)00398-8, 2001a.
  - Mader, B. T., and Pankow, J. F.: Gas/solid partitioning of semivolatile organic compounds (SOCs) to air filters. 3. An analysis of gas adsorption artifacts in measurements of atmospheric SOCs and organic carbon (OC) when using Teflon membrane filters and quartz fiber filters, Environ. Sci.

Technol., 35(17), 3422-3432, https://doi.org/10.1021/es0015951, 2001b.

- Malm, W. C., Molenar, J. V., Eldred, R. A., and Sisler, J. F.: Examining the relationship among 693 atmospheric aerosols and light scattering and extinction in the Grand Canyon area, J. Geophys. 694 Res.:Atmos., 101(D14), 19251-19265, https://doi.org/10.1029/96JD00552, 1996.
  - Ming, Y., Ramaswamy, V., Ginoux, P. A., and Horowitz, L. H.: Direct radiative forcing of anthropogenic organic aerosol, J. Geophys. Res.:Atmos., 110(D20), D20208, https://doi.org/10.1029/2004JD005573, 2005.
  - Ng, N., Kwan, A., Surratt, J., Chan, A., Chhabra, P., Sorooshian, A., Pye, H. O., Crounse, J., Wennberg, P., and Flagan, R.: Secondary organic aerosol (SOA) formation from reaction of isoprene with nitrate radicals (NO<sub>3</sub>), Atmos. Chem. Phys., 8(14), 4117-4140, https://doi.org/10.5194/acp-8-4117-2008, 2008.
    - Novakov, T., and Penner, J.: Large contribution of organic aerosols to cloud-condensation-nuclei concentrations, Nature, 365(6449), 823-826, http://doi.org/10.1038/365823a0, 1993.
    - Noziere, B., Kalberer, M., Claeys, M., Allan, J., D'Anna, B., Decesari, S., Finessi, E., Glasius, M., Grgic, I., and Hamilton, J. F.: The molecular identification of organic compounds in the atmosphere: state of the art and challenges, Chem. Rev., 115(10), 3919-3983, https://doi.org/10.1021/cr5003485, 2015.
    - Paatero, P., and Hopke, P. K.: Discarding or downweighting high-noise variables in factor analytic models, Anal. Chim. Acta., 490(1-2), 277-289, https://doi.org/10.1016/S0003-2670(02)01643-4, 2003.
    - Pankow, J. F.: An absorption model of gas/particle partitioning of organic compounds in the atmosphere, Atmos. Environ., 28(2), 185-188, https://doi.org/10.1016/1352-2310(94)90093-0, 1994a.
    - Pankow, J. F.: An absorption model of the gas/aerosol partitioning involved in the formation of secondary organic aerosol, Atmos. Environ., 28(2), 189-193, https://doi.org/10.1016/1352-2310(94)90094-9, 1994b.
    - Pankow, J. F., Storey, J. M., and Yamasaki, H.: Effects of relative humidity on gas/particle partitioning of semivolatile organic compounds to urban particulate matter, Environ. Sci. Technol., 27(10), 2220-2226, https://doi.org/10.1021/es00047a032, 1993.
    - Peters, A., Lane, D., Gundel, L., Northcott, G. L., and Jones, K. C.: A comparison of high volume and diffusion denuder samplers for measuring semivolatile organic compounds in the atmosphere, Environ. Sci. Technol., 34(23), 5001-5006, https://doi.org/10.1021/es000056t, 2000.
    - Pierson, W. R., Brachaczek, W. W., Korniski, T. J., Truex, T. J., and Butler, J. W.: Artifact formation of sulfate, nitrate, and hydrogen ion on backup filters: Allegheny Mountain experiment, J. Air Pollut. Control Assoc., 30(1), 30-34, https://doi.org/10.1080/00022470.1980.10465910, 1980.
    - Qin, C., Gou, Y., Wang, Y., Mao, Y., Liao, H., Wang, Q. g., and Xie, M.: Gas-particle partitioning of polyol tracers at a suburban site in Nanjing, east China: increased partitioning to the particle phase, Atmos. Chem. Phys., 21(15), 12141-12153, https://doi.org/10.5194/acp-21-12141-2021, 2021.
    - Quinn, T. R., Canham, C. D., Weathers, K. C., and Goodale, C. L.: Increased tree carbon storage in response to nitrogen deposition in the US, Nat. Geosci., 3(1), 13-17, https://doi.org/10.1038/ngeo721, 2010.
    - Shen, F., Zhang, L., Jiang, L., Tang, M., Gai, X., Chen, M., and Ge, X.: Temporal variations of six ambient criteria air pollutants from 2015 to 2018, their spatial distributions, health risks and relationships with socioeconomic factors during 2018 in China, Environ. Int., 137, 105556, https://doi.org/10.1016/j.envint.2020.105556, 2020.
    - Shen, H., Chen, Z., Li, H., Qian, X., Qin, X., and Shi, W.: Gas-particle partitioning of carbonyl compounds in the ambient atmosphere, Environ. Sci. Technol., 52(19), 10997-11006, https://doi.org/10.1021/acs.est.8b01882, 2018.
    - Simoneit, B. R., Elias, V. O., Kobayashi, M., Kawamura, K., Rushdi, A. I., Medeiros, P. M., Rogge, W. F., and Didyk, B. M.: Sugars dominant water-soluble organic compounds in soils and characterization as tracers in atmospheric particulate matter, Environ. Sci. Technol., 38(22), 5939-5949, https://doi.org/10.1021/es0403099, 2004.
    - Subramanian, R., Khlystov, A. Y., Cabada, J. C., and Robinson, A. L.: Positive and negative artifacts in particulate organic carbon measurements with denuded and undenuded sampler configurations special issue of aerosol science and technology on findings from the fine particulate matter supersites program, Aerosol Sci. Technol., 38(S1), 27-48, https://doi.org/10.1080/02786820390229354, 2004.
- Surratt, J. D., Chan, A. W., Eddingsaas, N. C., Chan, M., Loza, C. L., Kwan, A. J., Hersey, S. P., Flagan,
   R. C., Wennberg, P. O., and Seinfeld, J. H.: Reactive intermediates revealed in secondary organic
   aerosol formation from isoprene, Proc. Natl. Acad. Sci. U.S.A., 107(15), 6640-6645,

https://doi.org/10.1073/pnas.0911114107, 2010.

- Surratt, J. D., Kroll, J. H., Kleindienst, T. E., Edney, E. O., Claeys, M., Sorooshian, A., Ng, N. L.,
   Offenberg, J. H., Lewandowski, M., and Jaoui, M.: Evidence for organosulfates in secondary
   organic aerosol, Environ. Sci. Technol., 41(2), 517-527, https://doi.org/10.1021/es062081q,
   2007a.
- Surratt, J. D., Lewandowski, M., Offenberg, J. H., Jaoui, M., Kleindienst, T. E., Edney, E. O., and Seinfeld, J. H.: Effect of acidity on secondary organic aerosol formation from isoprene, Environ.
  Sci. Technol., 41(15), 5363-5369, https://doi.org/10.1021/es0704176, 2007b.
  Surratt, J. D., Murphy, S. M., Kroll, J. H., Ng, N. L., Hildebrandt, L., Sorooshian, A., Szmigielski, R.,
  - Surratt, J. D., Murphy, S. M., Kroll, J. H., Ng, N. L., Hildebrandt, L., Sorooshian, A., Szmigielski, R., Vermeylen, R., Maenhaut, W., and Claeys, M.: Chemical composition of secondary organic aerosol formed from the photooxidation of isoprene, J. Phys. Chem. A, 110(31), 9665-9690, https://doi.org/10.1021/jp061734m, 2006.
  - Tao, S., and Lin, B.: Water soluble organic carbon and its measurement in soil and sediment, Water Res., 34(5), 1751-1755, https://doi.org/10.1016/S0043-1354(99)00324-3, 2000.
    - Watson, J. G., and Chow, J. C.: Comparison and evaluation of in situ and filter carbon measurements at the Fresno Supersite, J. Geophys. Res.:Atmos., 107(D21), ICC 3-1-ICC 3-15, https://doi.org/10.1029/2001JD000573, 2002.
    - Watson, J. G., Chow, J. C., Chen, L.-W. A., and Frank, N. H.: Methods to assess carbonaceous aerosol sampling artifacts for IMPROVE and other long-term networks, J. Air Waste Manage. Assoc., 59(8), 898-911, https://doi.org/10.3155/1047-3289.59.8.898, 2009.
    - Williams, B. J., Goldstein, A. H., Kreisberg, N. M., and Hering, S. V.: In situ measurements of gas/particle-phase transitions for atmospheric semivolatile organic compounds, Proc. Natl. Acad. Sci. U.S.A., 107(15), 6676-6681, https://doi.org/10.1073/pnas.0911858107, 2010.
    - Wilson, J. G., Kingham, S., Pearce, J., and Sturman, A. P.: A review of intraurban variations in particulate air pollution: Implications for epidemiological research, Atmos. Environ., 39(34), 6444-6462, https://doi.org/10.1016/j.atmosenv.2005.07.030, 2005.
    - Worton, D. R., Surratt, J. D., LaFranchi, B. W., Chan, A. W., Zhao, Y., Weber, R. J., Park, J.-H., Gilman, J. B., De Gouw, J., and Park, C.: Observational insights into aerosol formation from isoprene, Environ. Sci. Technol., 47(20), 11403-11413, https://doi.org/10.1021/es4011064, 2013.
    - Xie, M., Hannigan, M. P., and Barsanti, K. C.: Gas/particle partitioning of n-alkanes, PAHs and oxygenated PAHs in urban Denver, Atmos. Environ., 95, 355-362, https://doi.org/10.1016/j.atmosenv.2014.06.056, 2014a.
    - Xie, M., Hannigan, M. P., and Barsanti, K. C.: Gas/particle partitioning of 2-methyltetrols and levoglucosan at an urban site in Denver, Environ. Sci. Technol., 48(5), 2835-2842, https://doi.org/10.1021/es405356n, 2014b.
    - Xie, M., Hays, M. D., and Holder, A. L.: Light-absorbing organic carbon from prescribed and laboratory biomass burning and gasoline vehicle emissions, Sci. Rep., 7(1), 7318, https://doi.org/10.1038/s41598-017-06981-8, 2016.
    - Xie, M., Lu, X., Ding, F., Cui, W., Zhang, Y., and Feng, W.: Evaluating the influence of constant source profile presumption on PMF analysis of PM<sub>2.5</sub> by comparing long-and short-term hourly observation-based modeling, Environ. Pollut., 314, 120273, https://doi.org/10.1016/j.envpol.2022.120273, 2022a.
  - Xie, M., Peng, X., Shang, Y., Yang, L., Zhang, Y., Wang, Y., and Liao, H.: Collocated measurements of light-absorbing organic carbon in PM<sub>2.5</sub>: Observation uncertainty and organic tracer-based source apportionment, J. Geophys. Res.:Atmos., 127(5), e2021JD035874, https://doi.org/10.1029/2021JD035874, 2022b.
  - Xu, J., Chen, J., Shi, Y., Zhao, N., Qin, X., Yu, G., Liu, J., Lin, Y., Fu, Q., and Weber, R. J.: First continuous measurement of gaseous and particulate formic acid in a suburban area of East China: Seasonality and gas—particle partitioning, ACS Earth Space Chem., 4(2), 157-167, https://doi.org/10.1021/acsearthspacechem.9b00210, 2019.
  - Xu, L., Guo, H., Boyd, C. M., Klein, M., Bougiatioti, A., Cerully, K. M., Hite, J. R., Isaacman-VanWertz, G., Kreisberg, N. M., and Knote, C.: Effects of anthropogenic emissions on aerosol formation from isoprene and monoterpenes in the southeastern United States, Proc. Natl. Acad. Sci. U.S.A., 112(1), 37-42, https://doi.org/10.1073/pnas.1417609112, 2015.
- Yang, L., Shang, Y., Hannigan, M. P., Zhu, R., Wang, Q. g., Qin, C., and Xie, M.: Collocated speciation of PM<sub>2.5</sub> using tandem quartz filters in northern nanjing, China: Sampling artifacts and measurement uncertainty, Atmos. Environ., 246, 118066, https://doi.org/10.1016/j.atmosenv.2020.118066, 2021.
- Yatavelli, R., Stark, H., Thompson, S., Kimmel, J., Cubison, M., Day, D., Campuzano-Jost, P., Palm, B., Hodzic, A., and Thornton, J.: Semicontinuous measurements of gas-particle partitioning of

- organic acids in a ponderosa pine forest using a MOVI-HRToF-CIMS, Atmos. Chem. Phys., 812 14(3), 1527-1546, https://doi.org/10.5194/acp-14-1527-2014, 2014.
- Yli, J., Taina, Zardini, A. A., Eriksson, A. C., Hansen, A. M. K., Pagels, J. H., Swietlicki, E., 814 Svenningsson, B., Glasius, M., Worsnop, D. R., and Riipinen, I.: Volatility of organic aerosol: 815 Evaporation of ammonium sulfate/succinic acid aqueous solution droplets, Environ. Sci. 816 Technol., 47(21), 12123-12130, https://doi.org/10.1021/es401233c, 2013.

819






826














- Yu, Y., He, S., Wu, X., Zhang, C., Yao, Y., Liao, H., Wang, Q. g., and Xie, M.: PM<sub>2.5</sub> elements at an urban site in Yangtze River Delta, China: High time-resolved measurement and the application in 1089-1099, apportionment, Environ. Pollut., 253, https://doi.org/10.1016/j.envpol.2019.07.096, 2019.
- Zhang, Q., Jimenez, J. L., Canagaratna, M., Allan, J. D., Coe, H., Ulbrich, I., Alfarra, M., Takami, A., Middlebrook, A., and Sun, Y.: Ubiquity and dominance of oxygenated species in organic aerosols in anthropogenically-influenced Northern Hemisphere midlatitudes, Geophys. Res. Lett., 34(13), L13801, https://doi.org/10.1029/2007GL029979, 2007.
- Zhang, T., Claeys, M., Cachier, H., Dong, S., Wang, W., Maenhaut, W., and Liu, X.: Identification and estimation of the biomass burning contribution to Beijing aerosol using levoglucosan as a marker. Atmos. Environ.. 42(29), 7013-7021. https://doi.org/10.1016/j.atmosenv.2008.04.050, 2008.
- Zhang, Y., He, X., Wang, C., Wang, X., Song, L., Lu, Z., Bi, X., and Feng, Y.: Methods and applications for quantitative assessment of uncertainty in atmospheric particulate matter source profiles, Atmos. Environ., 338, 120815, https://doi.org/10.1016/j.atmosenv.2024.120815, 2024.
- Zhang, Y., Sheesley, R. J., Bae, M.-S., and Schauer, J. J.: Sensitivity of a molecular marker based positive matrix factorization model to the number of receptor observations, Atmos. Environ., 43(32), 4951-4958, https://doi.org/10.1016/j.atmosenv.2009.07.009, 2009.
- Zhao, Y., Kreisberg, N. M., Worton, D. R., Isaacman, G., Weber, R. J., Liu, S., Day, D. A., Russell, L. M., Markovic, M. Z., and VandenBoer, T. C.: Insights into secondary organic aerosol formation mechanisms from measured gas/particle partitioning of specific organic tracer compounds, Environ. Sci. Technol., 47(8), 3781-3787, https://doi.org/10.1021/es304587x, 2013a.
- Zhao, Y., Kreisberg, N. M., Worton, D. R., Teng, A. P., Hering, S. V., and Goldstein, A. H.: Development 840 of an in situ thermal desorption gas chromatography instrument for quantifying atmospheric organic compounds, Aerosol Sci. Technol., 258-266, 842 https://doi.org/10.1080/02786826.2012.747673, 2013b.

Table 1. Mean concentrations of WSOMMs (ng  $m^{-3}$ ) in  $Q_f$  samples from Sampler I and II.

| Species                                                   | Abbreviation                     | Sampler I           | Sampler II      | Means of collocated samples |  |  |  |
|-----------------------------------------------------------|----------------------------------|---------------------|-----------------|-----------------------------|--|--|--|
| Isoprene SOA tracers                                      |                                  |                     |                 |                             |  |  |  |
| 2-methylglyceric acid                                     | 2-MG                             | $4.39 \pm 3.29^{a}$ | $4.57 \pm 3.05$ | $4.48 \pm 3.15$             |  |  |  |
| 2-methylthreitol <sup>b</sup>                             | 2-MTH                            | $3.57 \pm 1.83$     | $3.82 \pm 1.93$ | $3.69 \pm 1.87$             |  |  |  |
| 2-methylerythritol                                        | 2-MEH                            | $9.20 \pm 5.13$     | $9.67 \pm 5.29$ | $9.43 \pm 5.18$             |  |  |  |
| 2-methyltetrols                                           | 2-MTs <sup>c</sup>               | $12.8 \pm 6.91$     | $13.5 \pm 7.16$ | $13.1 \pm 7.00$             |  |  |  |
| cis-2-methyl-1,3,4-<br>trihydroxy-1-butene <sup>b</sup>   | cis-MTHB                         | $3.47\pm3.15$       | $3.70\pm3.12$   | $3.58 \pm 3.12$             |  |  |  |
| 3-methyl-2,3,4-<br>trihydroxy-1-butene <sup>b</sup>       | MTHB                             | $2.14\pm1.82$       | $2.25\pm1.85$   | $2.19\pm1.83$               |  |  |  |
| trans-2-methyl-1,3,4-<br>trihydroxy-1-butene <sup>b</sup> | trans-MTHB                       | $10.0\pm10.5$       | $10.6\pm10.3$   | $10.3\pm10.4$               |  |  |  |
| C <sub>5</sub> -alkene triols                             | C <sub>5</sub> -ATs <sup>d</sup> | $15.2 \pm 14.9$     | $16.0 \pm 14.7$ | $15.6 \pm 14.7$             |  |  |  |
|                                                           |                                  | Dicarboxylic acid   |                 |                             |  |  |  |
| succinic acid                                             |                                  | $20.8 \pm 13.9$     | $22.6 \pm 15.4$ | $21.7 \pm 14.5$             |  |  |  |
| glutaric acid                                             |                                  | $8.31 \pm 5.56$     | $8.63 \pm 4.98$ | $8.47 \pm 5.20$             |  |  |  |
| adipic acid                                               |                                  | $5.93 \pm 3.45$     | $6.59 \pm 3.94$ | $6.26 \pm 3.60$             |  |  |  |
|                                                           | Bi                               | omass burning trace |                 |                             |  |  |  |
| galactosan                                                |                                  | $0.36 \pm 0.51$     | $0.42 \pm 0.63$ | $0.39 \pm 0.57$             |  |  |  |
| mannosan                                                  |                                  | $1.68 \pm 1.04$     | $1.78 \pm 1.18$ | $1.73 \pm 1.11$             |  |  |  |
| levoglucosan                                              |                                  | $21.5 \pm 19.4$     | $22.9 \pm 20.2$ | $22.2 \pm 19.8$             |  |  |  |
|                                                           |                                  | Saccharides         |                 |                             |  |  |  |
| fructose                                                  |                                  | $12.5 \pm 8.87$     | $13.6 \pm 8.99$ | $13.1 \pm 8.82$             |  |  |  |
| glucose                                                   |                                  | $9.29 \pm 8.41$     | $10.2 \pm 9.04$ | $9.75 \pm 8.65$             |  |  |  |
| sucrose                                                   |                                  | $28.0 \pm 32.8$     | $29.7 \pm 33.7$ | $28.9 \pm 33.2$             |  |  |  |
| lactose                                                   |                                  | $1.61 \pm 1.37$     | $1.69 \pm 1.42$ | $1.65 \pm 1.40$             |  |  |  |
| mannose                                                   |                                  | $0.70 \pm 0.61$     | $0.79 \pm 0.64$ | $0.75 \pm 0.62$             |  |  |  |
| Sugar alcohols                                            |                                  |                     |                 |                             |  |  |  |
| arabitol                                                  |                                  | $5.97 \pm 4.66$     | $6.66 \pm 4.51$ | $6.31 \pm 4.56$             |  |  |  |
| pinitol                                                   |                                  | $1.07 \pm 0.82$     | $1.15 \pm 0.85$ | $1.11 \pm 0.84$             |  |  |  |
| mannitol                                                  |                                  | $16.9 \pm 23.0$     | $18.8 \pm 23.4$ | $17.9 \pm 23.1$             |  |  |  |
| sorbitol                                                  |                                  | $1.00 \pm 0.77$     | $1.10 \pm 0.72$ | $1.05 \pm 0.74$             |  |  |  |
| inositol                                                  |                                  | $2.11 \pm 1.03$     | $2.25 \pm 1.09$ | $2.18 \pm 1.04$             |  |  |  |
| chiro inositol                                            |                                  | $0.43 \pm 0.41$     | $0.47 \pm 0.41$ | $0.45 \pm 0.41$             |  |  |  |

<sup>&</sup>lt;sup>a</sup> Standard deviation; <sup>b</sup> compounds were quantified using meso-erythritol as the surrogate, and other compounds were quantified using authentic standards; <sup>c</sup> sum of 2-MTH and 2-MEH; <sup>d</sup> sum of trans-MTHB, MTHB, and cis-MTHB.

Table 2. Mean concentrations (ng m $^{-3}$ ), B, and F% of isoprene SOA tracers and dicarboxylic acids based on the measurement results of filter samples from Sampler II.

| Species             | $Q_f$                | $Q_b$             | $Q_{bb}$          | В               | F%              |  |  |  |  |
|---------------------|----------------------|-------------------|-------------------|-----------------|-----------------|--|--|--|--|
|                     | Isoprene SOA tracers |                   |                   |                 |                 |  |  |  |  |
| 2-MG                | $4.57 \pm 3.05$      | $0.85 \pm 0.72$   | $0.20\pm0.13$     | $24.1\pm10.2$   | $81.7 \pm 9.98$ |  |  |  |  |
| 2-MTH               | $3.82 \pm 1.93$      | $0.62\pm0.52$     | $0.16\pm0.16$     | $21.4 \pm 11.9$ | $83.9 \pm 9.36$ |  |  |  |  |
| 2-MEH               | $9.66 \pm 5.29$      | $1.13 \pm 1.16$   | $0.57 \pm 0.70$   | $32.2\pm13.3$   | $86.1 \pm 10.1$ |  |  |  |  |
| 2-MTs               | $13.5 \pm 7.16$      | $1.74 \pm 1.63$   | $0.73 \pm 0.86$   | $28.1 \pm 13.1$ | $85.5 \pm 9.62$ |  |  |  |  |
| cis-MTHB            | $3.70 \pm 3.12$      | $0.053 \pm 0.051$ | $0.035 \pm 0.025$ | $42.4 \pm 13.3$ | $95.9 \pm 4.38$ |  |  |  |  |
| MTHB                | $2.25 \pm 1.85$      | $0.064 \pm 0.035$ | $0.030 \pm 0.016$ | $32.8\pm8.00$   | $92.7 \pm 6.71$ |  |  |  |  |
| trans-MTHB          | $10.6 \pm 10.3$      | $0.16 \pm 0.21$   | $0.099\pm0.074$   | $42.6 \pm 13.5$ | $95.2 \pm 6.49$ |  |  |  |  |
| C <sub>5</sub> -ATs | $16.0 \pm 14.7$      | $0.29 \pm 0.29$   | $0.17 \pm 0.11$   | $40.0\pm11.7$   | $94.9 \pm 5.80$ |  |  |  |  |
| Dicarboxylic acids  |                      |                   |                   |                 |                 |  |  |  |  |
| succinic acid       | $22.6 \pm 15.4$      | $6.17 \pm 3.76$   | $3.66 \pm 2.18$   | $39.7 \pm 11.2$ | $68.1 \pm 8.22$ |  |  |  |  |
| glutaric acid       | $8.63 \pm 4.98$      | $2.36 \pm 1.46$   | $1.18\pm0.38$     | $36.4 \pm 12.1$ | $69.4 \pm 7.39$ |  |  |  |  |
| adipic acid         | $6.59 \pm 3.94$      | $1.10\pm0.70$     | $0.68 \pm 0.39$   | $40.0\pm13.8$   | $77.6 \pm 6.51$ |  |  |  |  |

Table 3. Comparisons of the mean concentrations (ng m<sup>-3</sup>) of isoprene SOA tracers and dicarboxylic acids in the  $Q_b$  and  $(NH_4)_2SO_4$ -treated  $Q_{bb}$  samples from Sampler I and the collocated untreated samples from Sampler II.

| Species             | Sampler I       |                 |                 | Sampler II        |                 |                 |  |  |
|---------------------|-----------------|-----------------|-----------------|-------------------|-----------------|-----------------|--|--|
| Species             | $Q_b$           | $Q_{bb}$        | $Q_{bb}/Q_b$    | $Q_b$             | $Q_{bb}$        | $Q_{bb}/Q_b$    |  |  |
| Isoprene tracers    |                 |                 |                 |                   |                 |                 |  |  |
| 2-MG                | $0.71\pm0.73$   | $0.16\pm0.12$   | $0.38 \pm 0.25$ | $1.01\pm0.84$     | $0.24 \pm 0.15$ | $0.36 \pm 0.23$ |  |  |
| 2-MTH               | $0.54 \pm 0.54$ | $0.88 \pm 0.64$ | $2.40\pm1.82$   | $0.70 \pm 0.61$   | $0.19 \pm 0.20$ | $0.30 \pm 0.20$ |  |  |
| 2-MEH               | $1.19\pm1.29$   | $2.46\pm2.01$   | $3.00\pm1.90$   | $1.42\pm1.44$     | $0.68 \pm 0.89$ | $0.49 \pm 0.24$ |  |  |
| 2-MTs               | $1.73\pm1.78$   | $3.34 \pm 2.64$ | $0.59 \pm 0.48$ | $2.13\pm2.00$     | $0.87 \pm 1.08$ | $0.42\pm0.23$   |  |  |
| cis-MTHB            | $0.035\pm0.024$ | $1.05\pm0.90$   | $30.7\pm15.1$   | $0.061 \pm 0.064$ | $0.036\pm0.029$ | $0.83 \pm 0.54$ |  |  |
| MTHB                | $0.046\pm0.025$ | $0.62\pm0.64$   | $11.5\pm7.74$   | $0.067 \pm 0.039$ | $0.031\pm0.018$ | $0.50\pm0.19$   |  |  |
| trans-MTHB          | $0.10\pm0.080$  | $2.25\pm1.75$   | $21.9\pm10.4$   | $0.20\pm0.27$     | $0.10\pm0.078$  | $0.81 \pm 0.45$ |  |  |
| C <sub>5</sub> -ATs | $0.19 \pm 0.13$ | $3.92 \pm 3.25$ | $20.3 \pm 9.87$ | $0.33\pm0.37$     | $0.17 \pm 0.12$ | $0.72 \pm 0.38$ |  |  |
| Dicarboxylic acid   |                 |                 |                 |                   |                 |                 |  |  |
| succinic acid       | $7.07 \pm 3.86$ | $4.67\pm5.27$   | $0.75 \pm 0.68$ | $8.07 \pm 4.17$   | $4.18\pm2.76$   | $0.63 \pm 0.26$ |  |  |
| glutaric acid       | $2.51\pm1.55$   | $1.99\pm1.27$   | $1.06\pm0.72$   | $2.97 \pm 2.04$   | $1.57\pm1.13$   | $0.67 \pm 0.62$ |  |  |
| adipic acid         | $1.08\pm0.59$   | $1.23 \pm 1.29$ | $1.41\pm1.31$   | $1.29\pm0.74$     | $0.71 \pm 0.41$ | $0.64 \pm 0.31$ |  |  |

Table 4. Comparisons of the mean concentrations (ng m<sup>-3</sup>) of isoprene SOA tracers and dicarboxylic acids in the  $Q_b$  and KOH-treated  $Q_{bb}$  samples from Sampler I and the collocated untreated samples from Sampler II.

| Chaoine             | Sampler I         |                 |                 | Sampler II       |                  |                 |  |  |
|---------------------|-------------------|-----------------|-----------------|------------------|------------------|-----------------|--|--|
| Species             | $Q_b$             | $Q_{bb}$        | $Q_{bb}/Q_b$    | $Q_b$            | $Q_{bb}$         | $Q_{bb}/Q_b$    |  |  |
|                     | Isoprene tracers  |                 |                 |                  |                  |                 |  |  |
| 2-MG                | $1.74\pm1.35$     | $1.92\pm1.38$   | $1.92\pm1.84$   | $0.62\pm0.51$    | $0.14 \pm 0.061$ | $0.39 \pm 0.32$ |  |  |
| 2-MTH               | $0.46\pm0.47$     | $0.15 \pm 0.11$ | $0.92 \pm 0.95$ | $0.51 \pm 0.39$  | $0.13\pm0.11$    | $0.31 \pm 0.24$ |  |  |
| 2-MEH               | $0.93\pm1.00$     | $0.38 \pm 0.25$ | $0.81\pm1.00$   | $0.78 \pm 0.60$  | $0.43\pm0.37$    | $0.60 \pm 0.49$ |  |  |
| 2-MTs               | $1.39\pm1.44$     | $0.47 \pm 0.30$ | $0.84 \pm 0.94$ | $1.29\pm0.93$    | $0.56 \pm 0.48$  | $0.47 \pm 0.37$ |  |  |
| cis-MTHB            | $0.068 \pm 0.062$ | $0.070\pm0.090$ | $1.60\pm2.72$   | $0.038\pm0.020$  | $0.031\pm0.021$  | $0.80 \pm 0.25$ |  |  |
| MTHB                | $0.056 \pm 0.037$ | $0.024\pm0.027$ | $0.92\pm1.26$   | $0.054\pm0.032$  | $0.029\pm0.013$  | $0.75 \pm 0.74$ |  |  |
| trans-MTHB          | $0.10\pm0.12$     | $0.033\pm0.051$ | $0.92\pm1.12$   | $0.11 \pm 0.079$ | $0.091\pm0.076$  | $0.91 \pm 0.37$ |  |  |
| C <sub>5</sub> -ATs | $0.22\pm0.16$     | $0.12 \pm 0.15$ | $0.92\pm1.30$   | $0.21\pm0.12$    | $0.15\pm0.11$    | $0.82 \pm 0.35$ |  |  |
| Dicarboxylic acid   |                   |                 |                 |                  |                  |                 |  |  |
| succinic acid       | $16.0\pm11.4$     | $7.05 \pm 5.39$ | $0.62 \pm 0.63$ | $4.90\pm2.72$    | $3.08\pm1.18$    | $0.81 \pm 0.93$ |  |  |
| glutaric acid       | $3.81 \pm 3.34$   | $1.50\pm1.71$   | $0.43 \pm 0.39$ | $2.06\pm1.03$    | $1.14 \pm 0.34$  | $0.83 \pm 0.93$ |  |  |
| adipic acid         | $2.91 \pm 5.63$   | $1.16\pm1.20$   | $0.67\pm0.57$   | $1.60\pm2.37$    | $0.71 \pm 0.43$  | $1.08 \pm 1.62$ |  |  |

Figure 1. Location of the sampling site (a) and scheme of collocated sampling with three stacked quartz filters (b).

Figure 2. Comparisons of the concentrations of typical WSOMMs in collocated  $Q_f$  samples (the red dashed line represents y=x).

Figure 3. Comparisons of mean concentrations of 2-MTs and C<sub>5</sub>-ATs in (a)  $Q_f$ , (b)  $Q_b$ , and (c)  $Q_{bb}$ /PUF samples between this study and Qin et al. (2021).

Figure 4. Comparisons of particulate concentrations of isoprene SOA tracers and dicarboxylic acid before and after gaseous adsorption corrections in summer 2021 (N: nighttime; D: day time).