# Peer review of "Measurement Report: Collocated speciation and potential"

_EGUsphere, 2025_

## Author Response (AR1)

**Responses to reviewer 1**

This study employed side-by-side filter samplers to characterize quartz filter sampling of organic molecular markers. The focus was on quantifying measurement uncertainty for primary and secondary water-soluble organic molecular markers. Although the topic of artifacts in the quartz filter sampling of organic compounds has been studied for decades, this study is novel because it focuses on many compounds that have not been characterized in such studies before. The manuscript is well organized and the writing is generally good. However, there are substantial technical problems with the manuscript, and I do not think most of the issues can be addressed through a major revision. These are detailed below.

**Response:**

Thanks for the reviewer's comments, and we will reply these point by point.

(1) The authors confuse precision with uncertainty throughout the manuscript. These are not the same thing. Sources of uncertainty in a filter measurement of organics come from a combination of sampling and analytical factors (flow rate, filter recovery including extraction, positive and negative artifacts, contamination throughout the filter/sample handling process, and GC analysis). Most of these are neglected in the manuscript. There are many good resources that explain this concept, including: "Modeling Uncertainty in the Earth Sciences, First Edition. Jef Caers. 2011 John Wiley & Sons, Ltd ISBN: 978-1-119-99263-9"

**Response:**

Thanks for the reviewer's comments. As mentioned in the *Introduction* of the manuscript,

"The total uncertainty for the characterization of atmospheric composition is composed of the uncertainties in both sampling and chemical analysis, and can be directly determined by performing collocated sampling. This method has been applied to estimate the concentration uncertainties of bulk PM components (Dutton et al., 2009a; Yang et al., 2021; Xie et al., 2022b), but the duplicate-derived uncertainty for the characterization of WSOMMs has rarely been investigated." (page 4, lines 87 – 93)

In this study, measurement results of water-soluble organic molecular markers (WSOMMs) in the top filters ( $Q_f$ ) of two collocated samplers were used to estimate their uncertainties by calculating the standard deviation of paired differences ( $SD_{diff}$ ) and the average relative percent difference (ARPD), as described in detail in *section* 2.3.3. Positive sampling artifacts due to adsorption of gaseous WSOMMs or their precursors were evaluated using backup filters ( $Q_b$  and  $Q_{bb}$ ), which are analyzed and discussed separately in *sections* 3.2 and 3.3.

In most previous studies, integrated filter samples from a single sampler were used to characterize particulate WSOMMs without considering uncertainties (e.g., Fu et al.,

2015; Xie et al., 2016; Bikkina et al., 2017; Schneider et al., 2022), and positive sampling artifacts of WSOMMs have rarely been investigated.

Regarding the confusion between precision and uncertainty, we would like to clarify that the inconsistent use of these two terms was identified and corrected before the open discussion. The current revised version of the manuscript does not contain this issue.

(2) The authors quantitatively treat evaporation of semi-volatile WSOMMs from the front quartz filter (i.e., the negative artifact) but they are not able to assess positive artifacts that arise from vapor deposition to the front quartz filter.

**Response:**

We appreciate the reviewer's comments.

As mentioned in the *Introduction* (pages 4 – 5, lines 94 – 110), the adsorption of gaseous WSOMMs on filters can be considered a positive artifact ("blow-on" effect), while the re-evaporation of particulate WSOMMs is a negative artifact ("blow-off" effect; Peters et al., 2000). Using a backup quartz filter downstream of the PM-loaded quartz filter has been employed in many studies to correct for adsorption of gaseous organics (e.g., Subramanian et al., 2004; Xie et al., 2014b).

In this study, the use of two backup filters ( $Q_b$  and  $Q_{bb}$ ) demonstrated the adsorption of WSOMMs or their precursors on filter media. A new approach was introduced to estimate the positive sampling artifact of WSOMMs by calculating the breakthrough value, as discussed in the last two paragraphs of section 3.2.

"Since adsorption of gaseous WSOMMs on quartz filters is a potential source of artifacts when sampling particulate WSOMMs (Arhami et al., 2006), previous studies have adjusted the particulate concentrations of organic compounds by subtracting the amounts on  $Q_b$  samples from those on  $Q_f$  samples ( $[Q_f]$ – $[Q_b]$ ) (Mader and Pankow, 2000, 2001a, 2001b). In this approach, the amounts of gaseous organic compounds adsorbed in  $Q_f$  and  $Q_b$  samples are assumed to be equal, and evaporation of the particle phase is neglected. However,  $Q_b$  is exposed to lower concentrations of gaseous WSOMMs before  $Q_f$  reaches equilibrium with the air sample (Mader and Pankow, 2001b; Watson et al., 2009). Then, the  $[Q_f]$ – $[Q_b]$  method may lead to an overestimation of particulate concentrations unless the sampling time is long enough (Hart and Pankow, 1994; Subramanian et al., 2004).

In Sampler II of this study, a third bare quartz filter  $(Q_{bb})$  was added after  $Q_f$  and  $Q_b$ , and the B values given in Table 2 also reflect the relationship between the amounts of gaseous WSOMMs adsorbed on two consecutive quartz filters. As such, it is more appropriate to estimate the amounts of gaseous WSOMMs adsorbed on  $Q_f([Q_f^*])$  by assuming that the B value of  $Q_f$  and  $Q_b([Q_b]/([Q_f^*]+[Q_b]))$  is identical to that of  $Q_b$  and  $Q_{bb}$ . In this case, the artifact-corrected particulate concentrations of the WSOMMs can be calculated as  $[Q_f]-[Q_f^*]$ . As Figure 4 shows, the  $[Q_f]$ ,  $[Q_f]-[Q_b]$ , and  $[Q_f]-[Q_f^*]$  values of all six species have similar time series. However, except for  $C_5$ -ATs, the mean  $[Q_f]$  and  $[Q_f]-[Q_b]$  values of 2-MG, 2-MTs, and dicarboxylic acids are 33.8% – 78.1%

and 11.1% - 40.3% higher than that of  $[Q_f]-[Q_f^*]$ . Since the volatilization of particulate WSOMMs in  $Q_f$  samples was not known, the values of  $[Q_f]-[Q_f^*]$  can be regarded as a lower limit for filter-based measurements of particulate WSOMMs." (Pages 15-16, lines 361-383)

Therefore, we did not quantify the evaporation of WSOMMs from the front quartz filter, but focused on estimating the positive artifact due to adsorption of gaseous WSOMMs or their precursors.

(3) Overall, the sampling was limited. There were only 24 paired samples collected in total and the temperature range during sample collection was fairly narrow (Table S1). This is probably not enough samples to support the major conclusions that the study states. Ambient temperature will certainly affect filter sampling artifacts (positive and negative) so the translation of these results to other studies and/or locations is limited.

**Response:**

Thanks for the reviewer's comments.

In this study, twenty-four pairs of three-layered filter samples were collected using two collocated samplers, resulting in a total of 144 samples. The top filters loaded with  $PM_{2.5}$  ( $Q_f$ ) from the two samplers (24 pairs) were used to estimate the measurement uncertainty of particulate WSOMMs, but not to represent their levels for the entire summer or year. As shown in Table 1 and Figures 2 and S1, all species showed similar mean concentrations between paired  $Q_f$  samples with no significant difference (p = 0.55 - 0.96), and the uncertainty of particulate WSOMMs derived from collocated sampling and analysis was reported for the first time. These results are useful for future modeling and field studies on the atmospheric transport, transformation, and source apportionment of water-soluble organic aerosols. Although the sample size is not large, the relative uncertainty values of WSOMMs (5.89% – 19.9%) were comparable to those of  $PM_{2.5}$  bulk components (5.28% – 23.7%; Yang et al., 2021) and non-polar OMMs (12.4% – 27.7%; Feng et al., 2026) derived from collocated sampling and analysis for a whole year at the same sampling site.

Similarly, the paired  $Q_b$  (24 pairs) and  $Q_{bb}$  (24 pairs) samples were used to demonstrate the adsorption of gaseous WSOMMs or their precursors to the filter media and to evaluate the adsorption mechanism, but not to represent the level of positive sampling artifact in summer or throughout the year. Based on comparisons of measurement results between paired samples, we developed a new method to correct for the adsorption of gaseous WSOMMs on PM-loaded filter samples. It was inferred that  $(NH_4)_2SO_4$  on quartz filters can promote the heterogeneous formation of isoprene SOA products by reactive uptake of isoprene epoxydiols (IEPOX), and KOH can increase the adsorption of gaseous organic acids on quartz filters by neutralization reactions. Due to the influence of surface reactions, WSOMMs detected in  $Q_b$  and  $Q_{bb}$  associated with SOA sources may not indicate their existence in the gas phase, which differs from the adsorption mechanisms reported for non-polar organic components

(e.g., polycyclic aromatic hydrocarbons) in previous studies (e.g., Mader and Pankow, 2002; Xie et al., 2014a). The development of the correction method and the identification of the potential adsorption mechanism for WSOMMs are not dependent on ambient temperature.

Therefore, the sample size of this study did not affect the validity of the major conclusions.

(4) From the results in Table 1, it appears there are systematic differences between Sampler I and Sampler II. The average concentration of the Qf sample for every compound is higher in Sampler II than for Sampler I. A statistical test can confirm if the difference is statistically significant but it certainly appears so. This presents significant complication for these analyses.

**Response:**

In this study, a statistical test (Student's t test) was conducted to compare the mean values of WSOMMs in paired  $Q_f$  samples (N = 24), and no significant difference (p = 0.55 - 0.96) was observed for any WSOMM. As mentioned in *Section 3.1.1*:

"The mean concentrations of WSOMMs and bulk PM2.5 components in collocated Qf samples are summarized in Tables 1 and S2, respectively. Generally, all species showed similar mean concentrations between paired Qf samples with no significant difference (Student's t test, p = 0.55 - 0.96)." (page 10, lines 232 – 235)

As shown in Figures 1 and S1, the scatter data of individual WSOMMs mostly lie on the identity line or are distributed on both sides. The slightly higher mean values of  $Q_f$  samples from Sampler II may be due to a few data points above the identity line with high values (e.g., adipic acid). Furthermore, the uncertainty of WSOMMs was estimated based on measurement results of all paired  $Q_f$  samples (SDdiff and ARPD, Section 2.3.3), not the mean  $Q_f$  values of Sampler I and II.

Therefore, the measurement uncertainties of particulate WSOMMs should not be attributed to systematic differences between Sampler I and Sampler II.

(5) As mentioned above, this general topic has been studied extensively. There are many nuances that are not addressed in this manuscript (effects of temperature, humidity, organic concentration, and filter face velocity, to name a few) – it is recommended that the authors delve into this huge body of literature to better understand their measurements and more accurately discuss the interpretations of their results.

**Response:**

Thanks for the reviewer's suggestions.

Based on collocated sampling of ambient air using three-layer quartz filters, the

primary goal of this work is to evaluate the measurement uncertainty of particulate WSOMMs, illustrate the adsorption of gaseous WSOMMs or their precursors on filter media (positive sampling artifact), and demonstrate that the presence of some WSOMMs on backup filters ( $Q_b$  and  $Q_{bb}$ ) is caused by the reactive uptake of precursors rather than physical adsorption. Temperature, relative humidity, organic concentration, and filter face velocity are well-known factors affecting the physical adsorption of bulk OC and non-polar organic compounds (e.g., n-alkanes and PAHs; McDow and Huntzicker, 1990; Pankow, 1992; Xie et al., 2013; Xie et al., 2014a). However, the presence of WSOMMs, particularly secondary products of isoprene, on backup filters is not likely caused by their adsorption in the gas phase.

As mentioned in Section 3.3 (pages 16 – 17, lines 392 – 420), the mean concentrations of 2-methyltetrols (2-MTs) and C5-alkene triols (C5-ATs) in (NH4)2SO4treated  $Q_{bb}$  samples from Sampler I were  $3.34 \pm 2.64$  ng m-3 and  $3.92 \pm 3.25$  ng m-3, respectively, which were 2.83 and 22.1 times higher than those in untreated  $Q_{bb}$  samples from Sampler II. According to previous studies, (NH4)2SO4 can act as an acid to promote the hydrolysis of IEPOX to form 2-MTs and C5-ATs or react with gaseous IEPOX as a nucleophile to form organosulfate esters and oligomeric forms of 2-MTs and C5-ATs (Surratt et al., 2006, 2010). During the derivatization process of sample analysis, the organosulfate and oligomeric forms of 2-MTs and C5-ATs can be converted to their monomeric forms by excess BSTFA (Lin et al., 2013; Xie et al., 2014b); the conventional GC/EI-MS method also overestimates the concentrations of 2-MTs and C5-ATs due to the thermal decomposition of less volatile oligomers and organosulfates (Lopez et al., 2016; Cui et al., 2018). In addition, Qb and untreated Qbb samples from Sampler II also contain a certain amount of inorganic sulfate due to heterogeneous reactions of SO2 (Pierson et al., 1980; Cheng et al., 2012), which are favored by the reactive uptake of IEPOX. Therefore, 2-MTs and C5-ATs detected in the Qb and Qbb samples from both Sampler I and II were likely generated by heterogeneous reactions of gaseous IEPOX on quartz filter surfaces rather than by direct adsorption of gaseous molecules. Because reactive uptake can significantly increase the adsorption of gaseous WSOMMs or their precursors, we propose developing chemically treated adsorbents for sampling in future studies.

The effects of temperature, humidity, organic concentration, and filter face velocity on adsorption are outside the scope of this work and cannot be comprehensively explained due to the limited sample size. We tentatively added some discussion on the relationships between the fractions of adsorbed WSOMMs in  $Q_b$  and  $Q_{bb}$  samples from Sampler II (A% = 1 - F%) and possible impacting factors, including temperature, relative humidity, OC concentration, and vapor pressure of individual WSOMMs. McDow and Huntzicker (1990) found that the amount of adsorbed OC decreased significantly with increasing filter face velocity. In this work and our previous study by Qin et al. (2021), the same samplers were used with a fixed face velocity of 25.3 cm s-1. Thus, the impact of filter face velocity was not discussed.

"According to the equilibrium G-P partitioning theory, a greater fraction of SVOCs exists in the gas phase when temperature rises (Pankow, 1994a, 1994b), as the vapor pressure of SVOCs increases exponentially with temperature. More adsorption sites on filter surfaces can be blocked by H2O molecules with increased RH, leading to lower adsorption of SVOCs (Pankow et al., 1993). Since absorption by particulate organic matter (OM) is an important G-P partitioning mechanism for ambient SVOCs (Liang and Pankow, 1996; Liang et al., 1997), increased OC concentrations might correspond to higher particle-phase fractions of SVOCs. However, the A% values of isoprene SOA tracers and dicarboxylic acids show little dependence on temperature, RH, and OC concentrations (Figures S4-S6), indicating more complex mechanisms for the adsorption of WSOMMs than for non-polar SVOCs (e.g., n-alkanes and PAHs). For example, due to the hygroscopicity of WSOMMs, water molecules attached to filter surfaces can promote gaseous adsorption. Dissolution in aerosol liquid water is more important than absorption by particulate OM for the equilibrium between particle- and gas-phase WSOMMs (Kampf et al., 2013; Isaacman et al., 2016; Shen et al., 2018; Qin et al., 2021). Although the mean A% values of dicarboxylic acids increased with their subcooled liquid vapor pressure  $(p^{o,*}_{L}; \text{ Figure S7})$ , this dependence was not observed when isoprene SOA tracers were included, which is assumed to result from their formation through heterogeneous reactions on filter surfaces." (pages 14 – 15, lines 341 -360)

Figure S4. Linear regressions between A% of individual WSOMMs and temperature.

Figure S5. Linear regressions between A% of individual WSOMMs and relative humidity (RH, %).

Figure S6. Linear regressions between A% of individual WSOMMs and OC concentrations.

Figure S7. Changes in mean A% with the vapor pressure of individual WSOMMs. The  $p^{o,*}_{L}$  data are obtained from Booth et al. (2010), Nguyen et al. (2011), and Qin et al. (2021).

(6) Section 3.1.1 is off topic from the rest of the manuscript.

**Response:**

As mentioned in our response to the reviewer's fifth comment, based on collocated sampling of ambient air using three-layer quartz filters, the primary goal of this work is to evaluate the measurement uncertainty of particulate WSOMMs, illustrate the adsorption of gaseous WSOMMs or their precursors on filter media (positive sampling artifact), and demonstrate that the presence of some WSOMMs on backup filters is caused by the reactive uptake of precursors rather than physical adsorption.

In Section 3.1, before discussing the measurement uncertainty of particulate WSOMMs (Section 3.1.2), an overview of the general agreement between collocated measurements and mean concentrations of particulate WSOMMs (Section 3.1.1) is important to demonstrate the validity of our results, which is necessary and cannot be omitted.

The reviewer's fourth comment also questioned the agreement between collocated measurements and requested the results of a statistical test, which were included in *Section 3.1.1* of the original manuscript.

**Reference:**

[revised manuscript text omitted]

This manuscript used stacked quartz filters to quantify the sampling uncertainties of WSOMMs. The paper provides useful information on many types of compounds and is valuable for the aerosol community. However, there are several issues that should be addressed before the manuscript can be considered for publication.

**Response:**

Thanks for the reviewer's comments. We will address these points one by one.

(1) The authors compared their WSOMM concentrations with previous studies. While useful, these comparisons are limited as they were conducted in different years under potentially different meteorological conditions. If the authors could also compare their quartz filter results to simultaneous measurements using other techniques, e.g. Teflon filters or online instrument, if such data are available.

**Response:**

The comparison of measurement results between this study and Qin et al. (2021) was conducted because both studies used the same sampler at the same sampling site. However, in this work, a quartz filter was used instead of a polyurethane foam (PUF) plug as the adsorbent for sampling gaseous WSOMMs. Although the two studies were conducted in different years, the particulate fractions (F%) of WSOMMs were compared for the same period (summer, August – September) under similar temperature and relative humidity conditions (Figure S3). The primary goal of this comparison is to show that quartz filters can be used instead of PUF, as using PUF did not result in a very different F% for isoprene SOA tracers in summer, and PUF sample analysis has low recoveries. This cannot be illustrated by comparisons between this study and previous studies that used other techniques (e.g., online measurements) at different sampling sites or during different time periods.

Teflon filters are rarely used for measuring organic compounds because they cannot be baked at high temperatures (e.g., > 450 °C) to eliminate background contamination.

Very few laboratories can perform online measurements for both gas- and particlephase WSOMMs, as mentioned in the *Introduction*:

"The Semi-Volatile Thermal Desorption Aerosol Gas chromatograph (SV-TAG) was developed for hourly measurements of WSOMMs in the gas and particle phase. In the SV-TAG, a parallel thermal desorption cell equipped with passivated high-surface-area stainless steel (SS) fiber filters (F-CTD) was used for sampling (Williams et al., 2010; Zhao et al., 2013a; 2013b; Isaacman et al., 2014, 2016). One F-CTD was used to directly collect WSOMMs in both the gas and particle phases, while the other cell was

set up to collect only WSOMMs in the particle phase by passing the sample air through an upstream activated carbon denuder. Comparisons between the two cells directly reflected the G-P partitioning of the WSOMMs. However, the resulting particulate fraction (F%) was often greater than 100% (Isaacman et al., 2016; Liang et al., 2023), possibly due to the uncertainties associated with the small sampling volume and chemical analysis." (Pages 5 – 6, lines 123 – 134)

Although the SV-TAG technique was developed for continuous measurements of gas- and particle-phase SVOCs (including WSOMMs), it was not commercially available and was almost always used by the same group of researchers. As mentioned above, the measurement results of the SV-TAG technique may be subject to large uncertainties.

Therefore, we did not include any comparison between this study and previous studies that used other techniques at different sampling sites or during different time periods.

(2) Besides temperature and relative humidities, particle loadings can be another factor affecting the gas-particle partitioning. The sampling campaign was conducted in summer. In winter, enhanced partitioning into the particle phase would be expected due to higher PM concentrations and lower temperatures. It would strengthen the manuscript if the authors can provide more discussions on how their findings may differ under such conditions, and to what extent the conclusions can be generalized.

**Response:**

As shown in Figure S3b, the mean  $PM_{2.5}$  concentration during the sampling periods in this study ( $15.3 \pm 5.29 \,\mu g \,m^{-3}$ ) was much lower than that in Qin et al. (2021) ( $28.2 \pm 10.3 \,\mu g \,m^{-3}$ ), but the F% of 2-methylterols (2-MTs) and  $C_5$ -alkene triols ( $C_5$ -ATs) were similar (Figure S3a). Moreover, F% values of 2-MTs and  $C_5$ -ATs did not show strong dependence on  $PM_{2.5}$  concentrations in either study (Figure R2-1).  $PM_{2.5}$  concentration data were obtained from a nearby online monitoring station in the same manner as Yu et al. (2019). Since dissolution in aerosol liquid water is more important than absorption by particulate OM for the equilibrium between particle- and gas-phase WSOMMs (Kampf et al., 2013; Isaacman et al., 2016; Shen et al., 2018; Qin et al., 2021), particle loadings may not be an important factor impacting the gas-particle partitioning of WSOMMs. Moreover, isoprene is mainly emitted from deciduous trees during the growing season, and its SOA products are rarely detected in winter.

Figure S3. Comparisons of F% of 2-MTs and C5-ATs and atmospheric conditions between this study and Qin et al. (2021).

Figure R2-1 Linear regressions between *F*% of C5-ATs and 2-MTs and PM2.5 concentrations.

In the revised manuscript, we added the mean  $PM_{2.5}$  concentrations in Figure S3(b) and expanded the discussion as follows:

"Based on PM2.5 data obtained from a nearby monitoring station using the same method as Yu et al. (2019), the mean PM2.5 concentration during the sampling period in Qin et al. (2021) (28.2  $\pm$  10.3  $\mu g$  m-3) was significantly (p < 0.05) higher than in this study (15.3  $\pm$  5.29  $\mu g$  m-3; Figure S3b), indicating that particle loading may not be a major factor affecting the G-P partitioning of isoprene SOA tracers." (Pages 13 – 14, lines 323 – 327)

"According to the equilibrium G-P partitioning theory, a greater fraction of SVOCs exists in the gas phase when temperature rises (Pankow, 1994a, 1994b), as the vapor pressure of SVOCs increases exponentially with temperature. More adsorption sites on

filter surfaces can be blocked by H2O molecules with increased RH, leading to lower adsorption of SVOCs (Pankow et al., 1993). Since absorption by particulate organic matter (OM) is an important G-P partitioning mechanism for ambient SVOCs (Liang and Pankow, 1996; Liang et al., 1997), increased OC concentrations might correspond to higher particle-phase fractions of SVOCs. However, the *A*% values of isoprene SOA tracers and dicarboxylic acids show little dependence on temperature, RH, and OC concentrations (Figures S4–S6), indicating more complex mechanisms for the adsorption of WSOMMs than for non-polar SVOCs (e.g., n-alkanes and PAHs). For example, due to the hygroscopicity of WSOMMs, water molecules attached to filter surfaces can promote gaseous adsorption. Dissolution in aerosol liquid water is more important than absorption by particulate OM for the equilibrium between particle- and gas-phase WSOMMs (Kampf et al., 2013; Isaacman et al., 2016; Shen et al., 2018; Qin et al., 2021)." (Pages 14 – 15, lines 341 – 356)

(3) For figure 2, can the authors describe the difference between black and red lines? Clear labeling in the figure caption would make it easier for readers to under the figure.

**Response:**

Thank you for the suggestion. In the captions of Figures 2 and S1 in the revised manuscript and supporting information, we added: "(the red dashed line represents y=x)".

(4) The authors only briefly mentioned that PM2.5 was collected. It is recommended to add more details of the collecting methods, including whether a size-selective inlet or cyclone was used, and what the sampling velocity or cut-off characteristics were.

**Response:**

The samplers have been used and described in several of our previous studies (e.g., Qin et al., 2021; Yang et al., 2021). In the revised manuscript, we have added the size-selective inlet type and face velocity.

"Three quartz filters (20.3 cm  $\times$  12.6 cm, Munktell Filter AB, Sweden) were stacked and placed on each of the two identical samplers (Sampler I and II; Mingye Environmental, Guangzhou, China) equipped with 2.5  $\mu$ m cut impactors to collect ambient air at a flow rate of 300 L min-1, with a filter face velocity of 25.2 cm S-1." (page 7, lines 156 – 159)


**Factor 1**, vapor pressure. The author already have the concentrations of many WSOMMs in different filter. The WSOMMs can be classified into two groups: low-volatility and high-volatility compound. Then a comprehensive vapor pressure analysis can be done to illustrate the impact of vapor pressure on WSOMMs adsorption.

**Factor 2**, humidity. What is impact of RH on WSOMMs with different structures (polar vs. non-polar compounds)? As the authors have already mentioned in the paper, the coated (NH4)2SO4 on Qbb can absorb water vapor and act as an acid to promote the hydrolysis of IEPOX on filters to form 2-MTs. This suggests that RH can also impact surface heterogeneous reactions. For a given WSOMM, did you observe different behaviors at different RH?

**Factor 3**, sampling flow rate. It would be valuable to know the impact of sampling flow rate, this can provide guidance for future sampling activity. More studies are also needed to address this issue, i.e., comparing the adsorption behaviors at low-flow rate and high-flow rate conditions.

**Response:**

As noted in our response to the first reviewer's fifth comment, based on collocated sampling of ambient air using three-layer quartz filters, the primary goal of this work is to evaluate the measurement uncertainty of particulate WSOMMs, illustrate the adsorption of gaseous WSOMMs or their precursors on filter media (positive sampling

artifact), and demonstrate that the presence of some WSOMMs on backup filters is caused by the reactive uptake of precursors rather than physical adsorption. Vapor pressure, relative humidity, and sampling flow rate are well-known factors affecting the physical adsorption of bulk OC and non-polar organic compounds (e.g., n-alkanes and PAHs; McDow and Huntzicker, 1990; Pankow, 1992; Xie et al., 2013; Xie et al., 2014a). However, the presence of WSOMMs, particularly secondary products of isoprene, on backup filters ( $Q_b$  and  $Q_{bb}$ ) is not likely caused by their adsorption in the gas phase.

As discussed in Section 3.3 (pages 16 – 17, lines 392 – 420), the mean concentrations of 2-methyltetrols (2-MTs) and C5-alkene triols (C5-ATs) in (NH4)2SO4treated  $Q_{bb}$  samples from Sampler I were  $3.34 \pm 2.64$  ng m-3 and  $3.92 \pm 3.25$  ng m-3, respectively, which were 2.83 and 22.1 times higher than those in untreated  $Q_{bb}$  samples from Sampler II. According to previous studies, (NH4)2SO4 can act as an acid to promote the hydrolysis of IEPOX to form 2-MTs and C5-ATs or react with gaseous IEPOX as a nucleophile to form organosulfate esters and oligomeric forms of 2-MTs and C5-ATs (Surratt et al., 2006; Surratt et al., 2010). During the derivatization process of sample analysis, the organosulfate and oligomeric forms of 2-MTs and C5-ATs can be converted to their monomeric forms by excess BSTFA (Lin et al., 2013; Xie et al., 2014b); the conventional GC/EI-MS method also overestimates the concentrations of 2-MTs and C5-ATs due to the thermal decomposition of less volatile oligomers and organosulfates (Lopez et al., 2016; Cui et al., 2018). In addition, Qb and untreated Qbb samples from Sampler II also contain a certain amount of inorganic sulfate due to heterogeneous reactions of SO2 (Pierson et al., 1980; Cheng et al., 2012), which are favored by the reactive uptake of IEPOX. Therefore, 2-MTs and C5-ATs detected in the  $Q_b$  and  $Q_{bb}$  samples from both Sampler I and II were likely generated by heterogeneous reactions of gaseous IEPOX on quartz filter surfaces rather than by direct adsorption of gaseous molecules. Because reactive uptake can significantly increase the adsorption of gaseous WSOMMs or their precursors, we propose developing chemically treated adsorbents for sampling in future studies.

The effects of vapor pressure, humidity, and sampling flow rate on adsorption are outside the scope of this work and cannot be comprehensively explained due to the limited sample size. We have tentatively added some discussion on the relationships between the fractions of adsorbed WSOMMs in  $Q_b$  and  $Q_{bb}$  samples from Sampler II (A% = 1 - F%) and possible impacting factors, including temperature, relative humidity, OC concentration, and vapor pressure of individual WSOMMs. McDow and Huntzicker (1990) found that the amount of adsorbed OC decreased significantly with increasing filter face velocity. In this work and our previous study by Qin et al. (2021), the same samplers were used with a fixed face velocity of 25.3 cm s-1. Thus, the impact of filter face velocity was not discussed.

"According to the equilibrium G-P partitioning theory, a greater fraction of SVOCs exists in the gas phase when temperature rises (Pankow, 1994a, 1994b), as the vapor pressure of SVOCs increases exponentially with temperature. More adsorption sites on

filter surfaces can be blocked by H2O molecules with increased RH, leading to lower adsorption of SVOCs (Pankow et al., 1993). Since absorption by particulate organic matter (OM) is an important G-P partitioning mechanism for ambient SVOCs (Liang and Pankow, 1996; Liang et al., 1997), increased OC concentrations might correspond to higher particle-phase fractions of SVOCs. However, the A% values of isoprene SOA tracers and dicarboxylic acids show little dependence on temperature, RH, and OC concentrations (Figures S4-S6), indicating more complex mechanisms for the adsorption of WSOMMs than for non-polar SVOCs (e.g., n-alkanes and PAHs). For example, due to the hygroscopicity of WSOMMs, water molecules attached to filter surfaces can promote gaseous adsorption. Dissolution in aerosol liquid water is more important than absorption by particulate OM for the equilibrium between particle- and gas-phase WSOMMs (Kampf et al., 2013; Isaacman et al., 2016; Shen et al., 2018; Qin et al., 2021). Although the mean A% values of dicarboxylic acids increased with their subcooled liquid vapor pressure  $(p^{o,*}_{L})$ ; Figure S7), this dependence was not observed when isoprene SOA tracers were included, which is assumed to result from their formation through heterogeneous reactions on filter surfaces." (pages 14 – 15, lines 341 -360)

Figure S4. Linear regressions between A% of individual WSOMMs and temperature.

Figure S5. Linear regressions between A% of individual WSOMMs and relative humidity (RH, %).

Figure S6. Linear regressions between A% of individual WSOMMs and OC concentrations.

Figure S7. Changes of mean A% with the vapor pressure of individual WSOMMs. The  $p^{o,*}_{L}$  data are obtained from Booth et al. (2010), Nguyen et al. (2011), and Qin et al. (2021).

**Minor comments**

4. Twenty-four pairs of samples were used for analysis. The sample number is relatively small.

**Response:**

As noted in our response to the first reviewer's third comment, in this study, twenty-four pairs of three-layered filter samples were collected using two collocated samplers, resulting in a total of 144 samples. The top filters loaded with PM2.5 ( $Q_f$ ) from the two samplers (24 pairs) were used to estimate the measurement uncertainty of particulate WSOMMs, but not to represent their levels for the entire summer or year. As shown in Table 1 and Figures 2 and S1, all species showed similar mean concentrations between paired  $Q_f$  samples with no significant difference (p = 0.55 - 0.96), and the uncertainty of particulate WSOMMs derived from collocated sampling and analysis was reported for the first time. These results are useful for future modeling and field studies on the atmospheric transport, transformation, and source apportionment of water-soluble organic aerosols. Although the sample size is not large, the relative uncertainty values of WSOMMs (5.89% – 19.9%) were comparable to those of PM2.5 bulk components (5.28% – 23.7%; Yang et al., 2021) and non-polar OMMs

(12.4% - 27.7%; Feng et al., 2026) derived from collocated sampling and analysis for a whole year at the same sampling site.

Similarly, the paired  $Q_b$  (24 pairs) and  $Q_{bb}$  (24 pairs) samples were used to demonstrate the adsorption of gaseous WSOMMs or their precursors to the filter media and to evaluate the adsorption mechanism, but not to represent the level of positive sampling artifact in summer or throughout the year. Based on comparisons of measurement results between paired samples, we developed a new method to correct for the adsorption of gaseous WSOMMs on PM-loaded filter samples. It was inferred that (NH4)2SO4 on quartz filters can promote the heterogeneous formation of isoprene SOA products by reactive uptake of isoprene epoxydiols (IEPOX), and KOH can increase the adsorption of gaseous organic acids on quartz filters by neutralization reactions. Due to the influence of surface reactions, WSOMMs detected in Qb and Qbb associated with SOA sources may not indicate their existence in the gas phase, which differs from the adsorption mechanisms reported for non-polar organic components (e.g., polycyclic aromatic hydrocarbons) in previous studies (Mader and Pankow, 2002; Xie et al., 2014a). The development of the correction method and the identification of the potential adsorption mechanism for WSOMMs are not dependent on sample size. Therefore, the sample size of this study did not affect the validity of our major conclusions.

5. For the treated filters, what are remaining concentrations of (NH4)2SO4 and KOH on different filters? The amount of (NH4)2SO4 and KOH on filters will significantly impact the adsorption behavior of certain WSOMMs.

**Response:**

As noted in our responses to previous comments, the comparison between chemically treated and untreated  $Q_{bb}$  samples is to show that the presence of some WSOMMs on backup filters results from the reactive uptake of precursors rather than physical adsorption. The results in Tables 3 and 4 indicate that  $(NH_4)_2SO_4$  on quartz filters can promote the heterogeneous formation of isoprene SOA products through the reactive uptake of isoprene epoxydiols (IEPOX), and that KOH can increase the adsorption of gaseous organic acids on quartz filters via neutralization reactions. Because reactive uptake can significantly increase the adsorption of gaseous WSOMMs or their precursors, we propose developing chemically treated adsorbents for sampling in future studies.

Due to the influence of surface reactions, WSOMMs detected in  $Q_b$  and  $Q_{bb}$  associated with SOA sources may not indicate their presence in the gas phase. As such, the comparison between chemically treated and untreated  $Q_{bb}$  samples is not intended to establish a relationship between the concentrations of  $(NH_4)_2SO_4$  and KOH on filters and adsorbed WSOMMs.

(NH4)2SO4 and KOH may not be evenly distributed on filters after soaking in solution and drying at 120°C. Additionally, the filter samples were folded in aluminum

foil after sampling to prevent the loss of collected substances, which also affected the distribution of (NH4)2SO4 and KOH on the filters. Thus, measuring the concentrations of (NH4)2SO4 and KOH in an aliquot of treated filters may not reflect their levels in the aliquots used for WSOMMs analysis.

Because measuring (NH4)2SO4 and KOH in treated filters is neither necessary nor appropriate for this study design, we did not measure their concentrations in treated filters.